# The phytochemical diversity of commercial *Cannabis* in the United States

**Christiana J. Smith[1], Daniela Vergara[2], Brian Keegan[3], Nick Jikomes [4]***

**1** Independent Researcher, Seattle, Washington, United States of America, **2** Department of Ecology and Evolutionary Biology, University of Colorado Boulder, Boulder, Colorado, United States of America, **3** Department of Information Science, University of Colorado Boulder, Boulder, Colorado, United States of America, **4** Department of Science and Innovation, Leafly Holdings Inc, Seattle, Washington, United States of America

\* njikomes@gmail.com

## Abstract

The legal status of *Cannabis* is changing, fueling an increasing diversity of *Cannabis*-derived products. Because *Cannabis* contains dozens of chemical compounds with potential psychoactive or medicinal effects, understanding this phytochemical diversity is crucial. The legal *Cannabis* industry heavily markets products to consumers based on widely used labeling systems purported to predict the effects of different "strains." We analyzed the cannabinoid and terpene content of commercial *Cannabis* samples across six US states, finding distinct chemical phenotypes (chemotypes) which are reliably present. By comparing the observed phytochemical diversity to the commercial labels commonly attached to *Cannabis*-derived product samples, we show that commercial labels do not consistently align with the observed chemical diversity. However, certain labels do show a biased association with specific chemotypes. These results have implications for the classification of commercial *Cannabis*, design of animal and human research, and regulation of consumer marketing—areas which today are often divorced from the chemical reality of the *Cannabis*-derived material they wish to represent.

## Introduction

*Cannabis sativa* L., a flowering plant from the family Cannabacea [1, 2], is one of the oldest domesticated plants [3]. It has been used by humans for more than 10,000 years [4] and has spread throughout the globe such that distinct varieties or cultivars exist today, each having been cultivated to express certain phenotypes. This versatile and phenotypically diverse plant has been used for a wide variety of commercial and medicinal purposes [1]. The *Cannabis* genus is considered to have a single species, *Cannabis sativa* L [5], inclusive of all forms of hemp and marijuana, with high genomic and phenotypic variation [6, 7] across multiple lineages [7–9]. 'Marijuana-type' lineages are used for human consumption (recreational and medical), while the 'hemp' lineages are used in industry settings for fiber or oil extraction.

For human consumption, the mature female inflorescences are grown, harvested, and processed into dried plant material. New laws leading to decriminalization and legalization have given rise to a global, multibillion dollar industry that is projected to continue to grow

**Data Availability Statement:** The final, cleaned dataset is available on the study's Github repository (https://github.com/cjsmith015/phytochemical-diversity-cannabis/) and contains all data and code necessary to run the analysis for all main figures,

as well as S2 and S4–S6 Figs. S1 and S3 Figs contain information specific to labs. The data underlying the results for S1 and S3 Figs are available upon request, with consent from each individual lab. The original raw data underlying the results presented in the study are available upon request, with consent from each individual lab that contributed data to the study before sharing their data. Contact information for data sharing inquiries are below. CannTest: Jonathan Rupp, jonathan@canntest.com ChemHistory: Patrick Trujillo, patrick@chemhistory.com Confidence Analytics: Nick Mosely, nick@conflabs.com Modern Canna Labs: George Fernandez, george@moderncanna.com PSI Labs: Ben Rosman, ben@psilabs.org SC Labs: Josh Wurzer, josh@sclabs.com Leafly: Nick Jikomes, nick.jikomes@leafly.com.

**Funding:** Leafly allowed N.J. to use some professional time to oversee this research project and work on the manuscript but did not provide other forms of funding. Leafly had no role in study design, lab data collection and analysis, decision to publish, or preparation of the manuscript.

**Competing interests:** I have read the journal's policy and the authors of this manuscript have the following competing interests: D.V. is the founder and president of the non-profit organization Agricultural Genomics Foundation, and the sole owner of CGRI, LLC. N.J. is employed by Leafly Holdings, Inc. Leafly allowed N.J. to use some professional time to oversee this research project and work on the manuscript.

substantially [10]. Beyond dried flowers, there are concentrated oils, beverages, topicals, suppositories, and other delivery mechanisms [11, 12]. To avoid confusion with the confounding terminology [13], we will use "*Cannabis*" in reference to the plant genus including its different varieties, "*Cannabis*-derived flower products" to refer to dried plant material produced for commercial purposes, and "other derivative products" to refer to other commercial form factors.

*Cannabis* is renowned for its secondary metabolites, including cannabinoids and terpenes. Cannabinoids are a class of compounds that can interact with the endocannabinoid system [14] and many have medicinal [15, 16] or psychoactive [3, 17] properties. Two of the most abundant cannabinoids are Δ-9-tetrahydrocannabinolic acid (THCA) and cannabidiolic acid (CBDA), which are converted to the neutral forms Δ-9-tetrahydrocannabinol (THC) and cannabidiol (CBD) once heated [18]. The enzymes that produce these cannabinoids are highly similar at the biochemical structure and genetic sequence levels [19, 20] and accept the same substrate, cannabigerolic acid (CBGA) [21, 22]. Although it was initially thought that CBDA and THCA were produced by one locus with two alleles [23], it is now known that these compounds are produced by multiple genes [24, 25] which vary in copy number [20] and allelic diversity [19]. It is therefore likely that the phenotypic expression of these genes corresponds to an additive effect of genes along with environmental factors (e.g. cultivation conditions).

Beyond THC and CBD, there are various cannabinoid compounds typically present at much lower levels. This includes CBGA, the precursor molecule to both THCA and CBDA. A third compound, CBCA (cannabichromenic acid). The ratios of these cannabinoids vary within and between varieties [20]. Other cannabinoids include cannabinol (CBN), a byproduct that accumulates with the breakdown of THC [26–28], Δ-9-tetrahydrocannabivarin carboxylic acid (THCVA), and others. Similar to THCA and CBDA, decarboxylation is responsible for the formation of cannabigerol (CBG), Δ-9-tetrahydrocannabivarin (THCV), and other neutral cannabinoids [29]. Due to their low abundance, these have generally been less well-studied than THC and CBD, although they display a range of interesting pharmacological properties with potential medicinal value [30–32].

Cannabinoid levels have been used both in setting legal definitions for different categories of cannabis products and for 'chemotaxonomic' purposes to classify different *Cannabis* cultivars based on THC:CBD ratios [33]. For example, the legal definition of hemp in the United States means, "the plant species *Cannabis sativa L.* and any part of that plant. . . with a Δ-9-tetrahydrocannabinol concentration of not more than 0.3% on a dry weight basis" (US Federal Registrar). This number intends to distinguish *Cannabis* with low intoxication potential from varieties containing high THC levels. Commercial marijuana-type *Cannabis* usually falls within discrete groups based on THC:CBD ratios [33], and has been categorized as either "THC-dominant" (low CBD levels), "CBD-dominant," (low THC levels and high CBD levels), or "Balanced THC/CBD" (comparable levels of THC and CBD), although the vast majority is THC-dominant [34]. The level of other minor cannabinoids has additionally been measured in a limited number of studies [35, 36]. However, a more comprehensive quantification of both major and minor cannabinoids from a large sample representative of commercial *Cannabis*, across multiple legal markets in the United States, is needed.

In addition to cannabinoids, *Cannabis* harbors a diverse class of related compounds known as terpenes [37, 38]. These are a type of secondary metabolite which often play defensive roles for the plant [39, 40]. They are responsible for its odors, can be pharmacologically active [17, 41], and may serve as reliable chemotaxonomic markers for classifying *Cannabis* beyond THC:CBD ratios [36, 42]. It has been shown that the chemical phenotype ("chemotype") of plants can be used to classify *Cannabis* into chemical varieties ("chemovars") [43, 44]. Distinct

chemovars, each with different ratios of cannabinoids and terpenes, are hypothesized to cause distinct effects for human consumers [44].

A variety of studies have looked at the chemical composition of *Cannabis* samples limited to a single geographic location [35, 36, 42, 43], included measurements of a limited number of cannabinoids [34, 45–49], or included measurements of terpenes without cannabinoid content [45]. Few studies have investigated the major and minor cannabinoids together with the terpenes [50] and none have performed a thorough chemotaxonomic analysis on a dataset with tens of thousands of samples across several regions of the United States. Mapping the chemical diversity of the *Cannabis*-derived products consumed by millions of people has important implications for consumer health and safety, such as identifying the number of chemically distinct types of *Cannabis* being consumed in legal markets. This may be consequential if distinct chemotypes are later determined to cause reliably different effects.

One source of variation among *Cannabis* is genetic differences that produce biochemical variation. Another source of is environmental conditions, including differences in light levels, soil nutrient composition, and other factors that can vary between cultivators [48]. The dataset we analyzed is composed of samples submitted by thousands of distinct cultivators across six US states. Because we cannot distinguish between genetic vs. environmental factors in driving any observed phytochemical variation, we focused on describing patterns of variation across regions, especially those that are consistent. We also quantified how well commercial product labels capture this chemical diversity.

It has been suggested that the multiple compounds produced by *Cannabis* may act in combination to produce specific medicinal and psychoactive effects, the so-called 'entourage effect' [15]. There is limited suggestive evidence for such an effect [41, 51], including improved patient outcomes in those who use whole-plant extracts (containing THC and unknown quantities of other compounds) versus synthetic THC [52]. For example, synthetic THC alone, in manufactured products such as 'Marinol,' may produce unpleasant effects [53, 54]. Whether or not distinct ratios of cannabinoids and terpenes are able to consistently yield different subjective effects or therapeutic outcomes is unknown, and a topic of debate [55].

Combinatorial effects, when the ingestion of two or more compounds yields different effects from either compound in isolation, are possible through various mechanisms [56–58], but largely unexplored for *Cannabis*-derived compounds beyond THC and CBD [59–62]. Carefully controlled *in vivo* studies are needed to determine whether distinct ratios of compounds have combinatorial effects. A first step toward defining possible chemical ratios to be used for *vivo* studies is to quantify the ratios present in commercial *Cannabis*-derived products. Doing so will also be important for informing the design of human clinical studies aimed at investigating the purported therapeutic effects of these products. Ideally, such studies will test formulations with cannabinoid and terpene ratios comparable to those that are widely encountered by millions of consumers today.

Another important reason to quantitatively map the chemotaxonomy of commercial *Cannabis* is that products are commonly labelled with distinct "strain names" or other categorical labels with alleged effects, implying that distinct chemical combinations are consistently linked to those labels. For example, consumers often believe that products labelled "Indica" are reliably sedating, while flower labelled as "Sativa" provide energizing effects [1, 7, 8]. *Cannabis*-derived products are aggressively marketed using these labels. A better understanding of whether these labels have any reliable association with distinct chemical profiles may have implications for consumer health and safety as well as the regulation of consumer marketing.

The lack of a standardized, regulated naming system for commercial *Cannabis* has been discussed previously [7, 9, 46]. Various studies, each limited in different ways, have investigated whether these labels capture real chemical variation. For example, cannabinoid and terpene

measurements from California found limited differences between "Indica" and "Sativa," with some "strain names" more consistently associated with specific chemical compositions than others [48]. Flower samples from the Netherlands were found to contain specific terpenes more often associated with "Indica" than to "Sativa" samples [49]. Samples from Washington state limited to total THC and CBD content found no differences between "Indica" and "Sativa," with potency variation between certain "strain names" [34]. Cannabinoid samples across the US did not find a clear relationship between "strain name" and chemotype, although terpene measurements were not included [46].

In this study, we conducted the largest chemotaxonomic analysis of commercial *Cannabis*-derived flower to date ($n$ = 89,923), using samples from testing labs in six US states. These samples were submitted by cultivators for testing in order to achieve compliance with state laws, representing *Cannabis*-derived products destined for sale in retail locations within each state. We analyzed both the cannabinoid and terpene content available for these samples, together with common industry labels and popularity metrics associated with them by the consumer-facing technology platform and data aggregator, Leafly. We defined distinct chemotypes that reliably show up across US states and quantified how well the industry labels "Indica," "Hybrid," and "Sativa" map to these chemotypes. We also examined the consistency of "strain names" across samples from different cultivators. These results provide new possibilities for systematically categorizing commercial *Cannabis* based on chemistry, the design of preclinical and clinical research experiments, and the regulation of commercial *Cannabis* marketing.

## Results

### Cannabinoid composition of U.S. commercial *Cannabis*

To assess total cannabinoid levels across samples, we plotted the distribution for each cannabinoid that was consistently measured across regions (Fig 1A) and for every cannabinoid measured within each region (S1 Fig). In all regions, total THC levels were much higher compared to levels of all other cannabinoids. Total CBD and CBG were present at modest levels in some samples, while other minor cannabinoids were usually present at very low levels (Figs 1A and S1). Following past work [34, 45], we established the presence of three distinct chemotypes based on THC: CBD ratios by plotting total THC against total CBD levels (Fig 1B; see Methods). Most samples belonged to the THC-dominant chemotype (96.5%) in the aggregate dataset (Fig 1B and 1C) and in each individual region (S2 Fig). A much smaller proportion of samples were classified as CBD-dominant (1.4%) or Balanced THC:CBD (2.2%; Figs 1 and S1). While we observed some regional differences (e.g. in mean total THC), we cannot determine how much of this reflects true variation in the regional samples vs. methodological differences between labs.

Although most samples contained low levels of cannabinoids beyond THC, we observed that 3.9% and 23.1% of samples, respectively, had total CBD or total CBG of 1% by weight or higher. To further understand any systematic patterns of variation in cannabinoid profiles beyond THC and CBD levels, we performed Principal Component Analysis (PCA) on all samples that contained measurements for total THC, CBD, CBG, CBC, CBN, and THCV content. Most of the variance in this dataset (96%) was explained by the first principal component (Fig 1D), which was highly correlated with samples' THC:CBD ratios ($r_s$ = -0.51, $P$ < 0.0001). Most of the remaining variation (3.6%) was explained by the second principal component, which was highly correlated with total CBG levels ($r_s$ = 0.95, $P$ < 0.0001). Thus, the vast majority of variance in these cannabinoid profiles is explained by variation among the three most abundant cannabinoids (THC, CBD, CBG) in commercial *Cannabis* in the US.

To further understand the relationship between each pair of these three cannabinoids, we plotted total levels of THC, CBD, and CBG against each other, separately for each THC:CBD

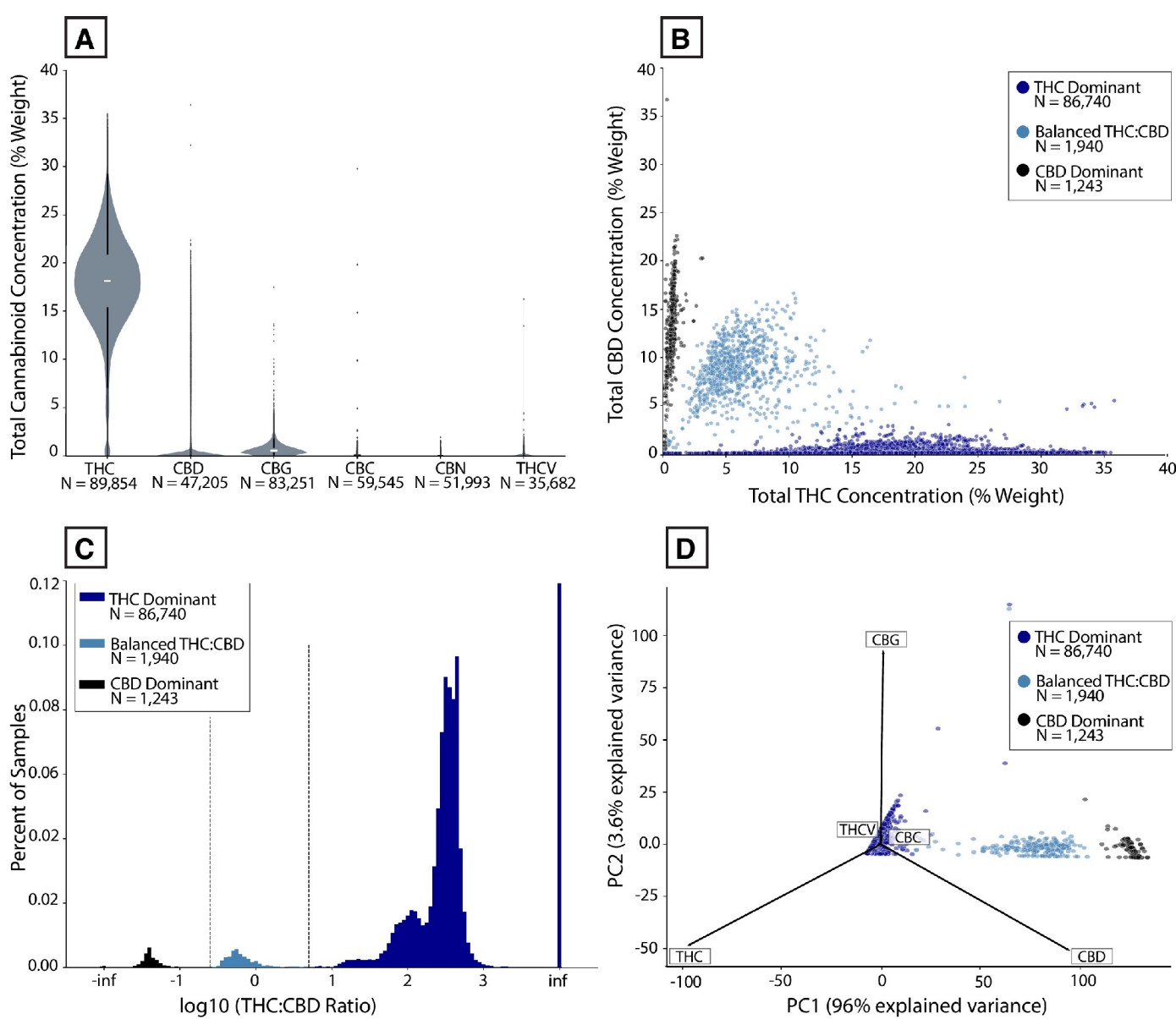

**Fig 1. Cannabinoid variation among commercial *Cannabis*-derived product samples in the US. (A)** Violin plot of distribution of the set of common cannabinoids measured across all regions. **(B)** Total THC vs. Total CBD levels, color-coded by THC:CBD chemotype. **(C)** Histogram showing THC:CBD distribution on a $\log_{10}$ scale. "Inf" stands for "infinite" (any samples with 0 total THC or CBD). **(D)** Principal Component Analysis of all cannabinoids shown in panel A, color-coded by THC:CBD chemotype.

chemotype. Given that CBGA is the precursor molecule to both THCA and CBDA, we expected to see positive correlations between each cannabinoid pair. This is what we observed, with the strength of each correlation varying across THC:CBD chemotypes (Fig 2). One notable finding with potential regulatory consequences is the substantial correlation between total THC and CBD levels in CBD-dominant samples ($r_s = 0.64$, $P < 0.0001$). 84.5% of CBD-dominant samples had total THC levels above 0.3%, the threshold used to legally define hemp in the US. This indicates that a substantial fraction of CBD-dominant *Cannabis* would not meet the legal definition of hemp in the US.

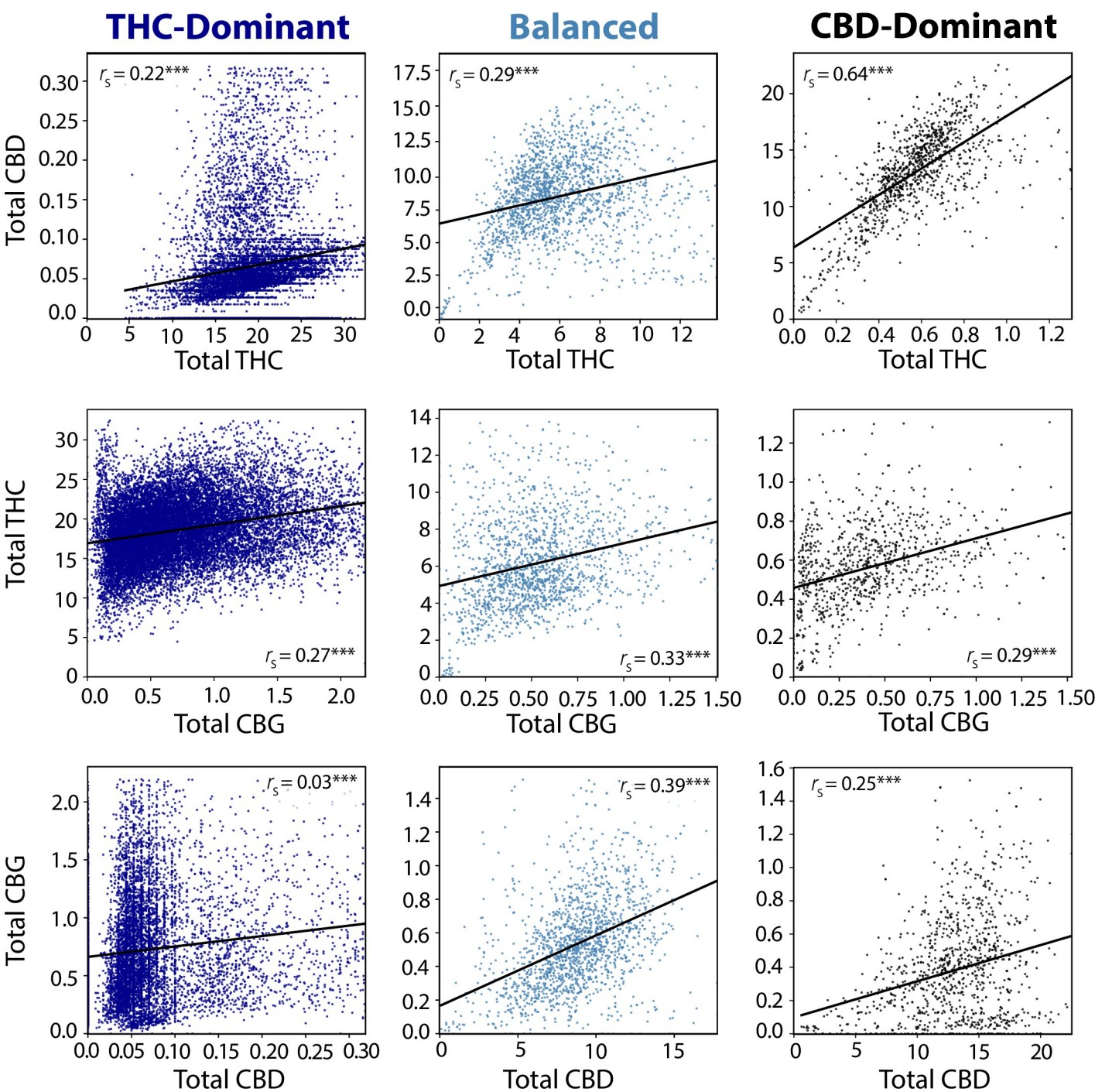

**Fig 2. Correlations among total THC, CBD, and CBG levels in each THC:CBD chemotype.** Scatterplots showing the linear correlation between total THC, CBD, and CBG levels in each of the main THC:CBD chemotypes. The sample sizes for each of the groups is as follows: THC-dominant N = 82,563; Balanced N = 1,876, and CBD-dominant N = 1,188. Top Row: Total THC vs. Total CBD; middle row: Total CBD vs. Total THC. Bottom row: Total CBD vs. Total CBG. ***$P < 0.0001$.

### Terpene composition of U.S. commercial *Cannabis*

We next assessed which terpene compounds were most prominent in samples by plotting the distribution of each terpene that was consistently measured in each region. On average, the

terpenes myrcene, β-caryophyllene, and limonene were present at the highest levels (Fig 3A). In most cases, individual terpenes were rarely present at more than 0.5% weight and most were present at low levels (< 0.2%) in a majority of samples. Overall, total terpene content averaged 2% by weight and displayed a modest but robust positive correlation with total cannabinoid content ($r_s$ = 0.37, $P$ < 0.0001), suggesting that the production of one type of compound doesn't come at the expense of the other.

To validate that we observe patterns expected from previous studies, we first looked for correlations between specific terpene pairs. We chose pairs that have been previously observed to display robust positive correlations, likely stemming from constraints on their biochemical synthesis [63–65]. Strong positive correlations were seen between α- and β-pinene (Fig 3B; $r_s$ =

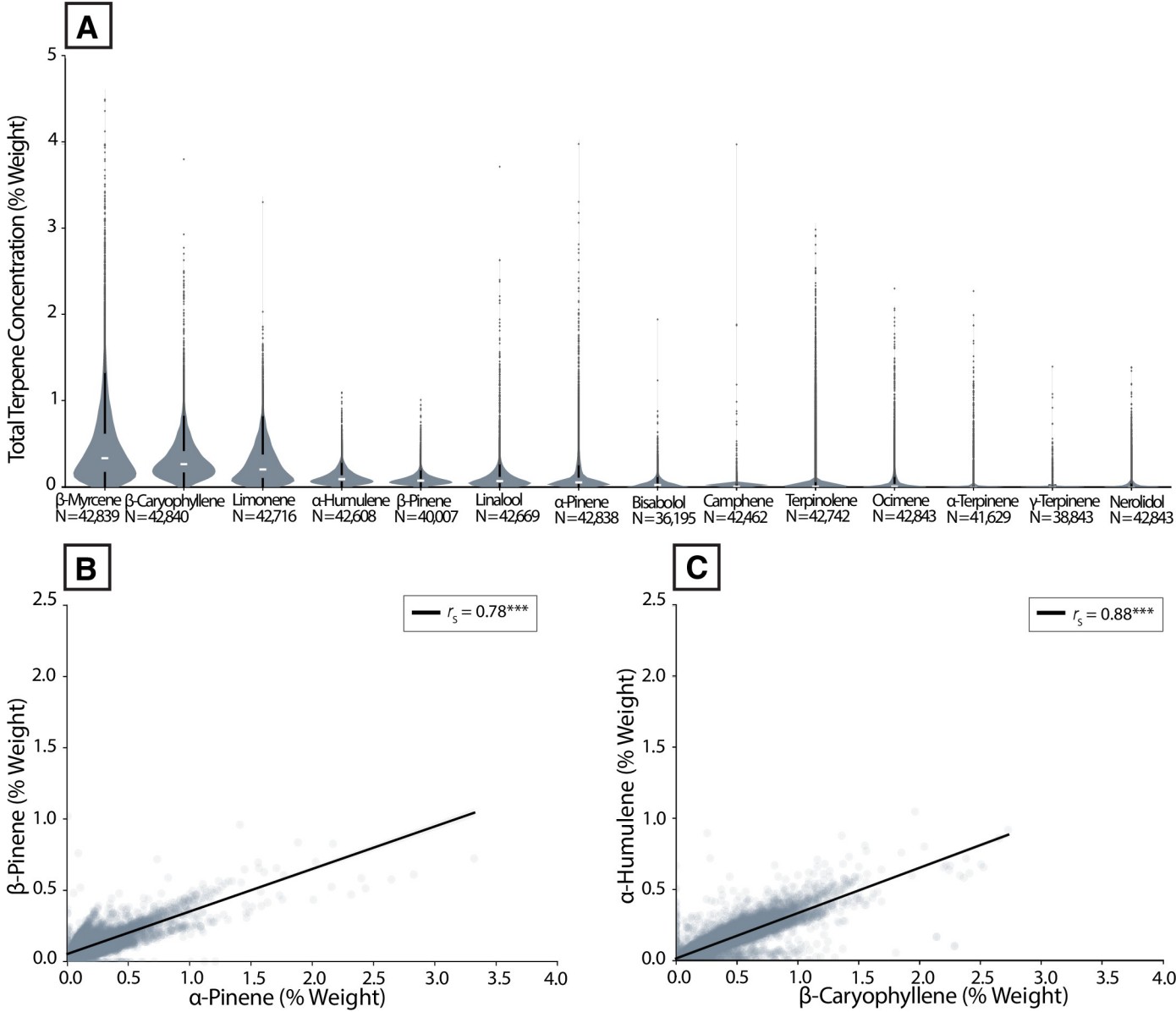

**Fig 3. Terpene abundance across commercial *Cannabis*-derived product samples in the US. (A)** Violin plots showing distributions of the set of common terpenes measured across all regions. **(B)** Scatterplot showing the correlation between α- and β-pinene, two common pinene isomers. $R_s$ = 0.78, \*\*\*$P$ < 0.0001 **(C)** Scatterplot showing the correlation between β-caryophyllene and humulene, two *Cannabis* terpenes co-produced by common enzymes. $R_s$ = 0.88, \*\*\*$P$ < 0.0001.

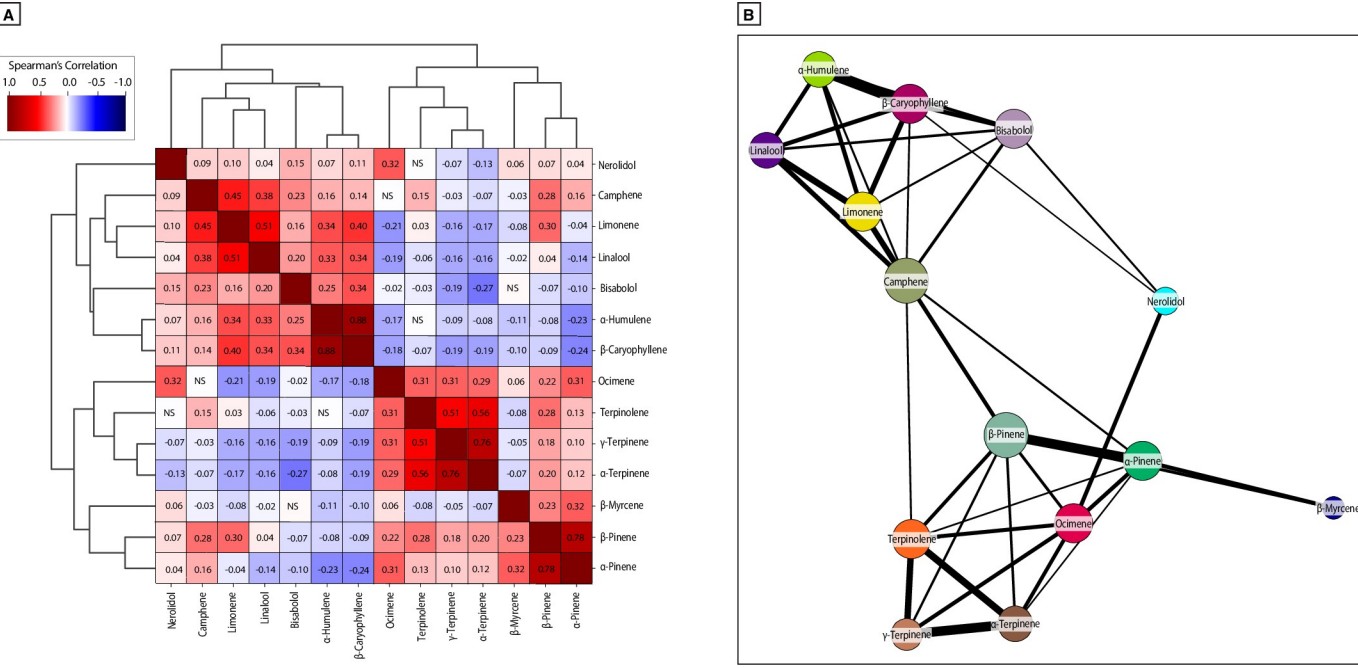

**Fig 4. Patterns of terpene co-occurrence among commercial *Cannabis*-derived product samples in the US. (A)** Hierarchically clustered correlation matrix showing pairwise correlations between all terpenes consistently measured across regions. **(B)** Network diagram where nodes are terpenes and edges are thresholded to the strongest observed correlations and their widths correspond to the strength of the correlation.

0.78, $P < 0.0001$), as well as β-caryophyllene and humulene (Fig 3C; $r_s = 0.88$, $P < 0.0001$). These correlations held for both the aggregate dataset (Fig 3) and for each individual US state (S3 and S4 Figs), demonstrating their robustness across regions.

To systematically understand relationships between all terpene pairs, we performed hierarchical clustering on all pairwise correlations among terpenes (Fig 4A; see Methods). This revealed distinct clusters of commonly co-occurring terpenes. After controlling for multiple comparisons, we observed many robust correlations between terpenes (see Methods). We also plotted this data in the form of a network diagram configured to display connections between terpenes with the strongest correlations (Fig 4B). This diagram provides a more compact picture of terpene co-occurrence and likely reflects the underlying biosynthesis pathways that give rise to these correlations [63–65].

## THC-dominant and high-CBD *Cannabis* display distinct levels of terpene diversity

Historically, the major focus of both clandestine and legal *Cannabis* breeding in the US has been on THC-dominant ("Type I") cultivars, which is why they predominate in the commercial marketplace (Fig 1) [2]. It is therefore expected that THC-dominant cultivars will display a more diverse array of terpene profiles than balanced THC:CBD ("Type II") and CBD-dominant ("Type III) cultivars. To visualize patterns of variation among terpene profiles, we performed a Principal Component Analysis (PCA; see Methods). The first three principal components explained 78.7% of the variance in the data (Fig 5A), indicating that most of the variance in terpene profiles can be explained with just a few components.

To visualize how patterns of terpene profile variation map to the major THC:CBD chemotypes shown in Fig 1, we plotted PCA scores for all samples along the first three principal

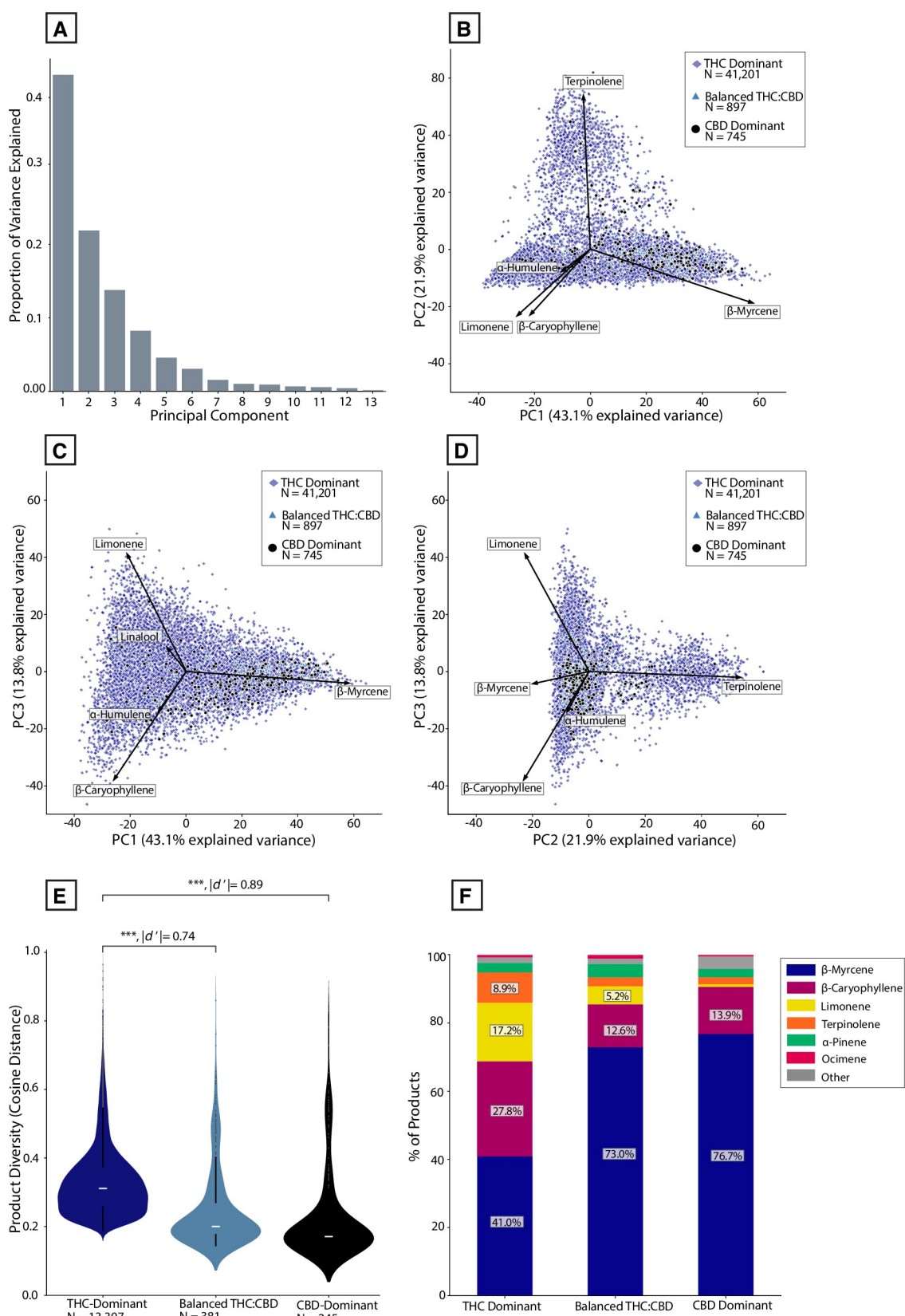

**Fig 5. Patterns of terpene profile diversity across THC:CBD chemotypes. (A)** Histogram showing the proportion of variation explained by each principal component after performing Principal Component Analysis on the terpene dataset. **(B)** PCA scores plotted along PC1 and PC2, color-coded by major THC:CBD chemotype. Vectors depict the loadings of the five individual terpenes onto these principal axes. **(C)** PCA scores plotted along PC1 and PC3. **(D)** PCA scores plotted along PC2 and PC3. **(E)** Violin plot showing distribution of 'product diversity' values (cosine distances) for each THC:CBD chemotype. Product values are calculated by averaging samples with the same strain name linked to a given producer ID. ***$P < 0.0001$, Welch's t-test and Cohen's d'. **(F)** Stacked bar chart showing the percent products with a given dominant terpene for each THC:CBD chemotype.

components, with each sample color-coded by its THC:CBD chemotype (Fig 5B–5D). The superimposed vectors encoding the five terpenes with the strongest loadings onto each principal component help clarify the terpene composition of different points on the graph. Most CBD-dominant and balanced THC:CBD samples cluster within a smaller subsection of the plots compared to THC-dominant samples. To quantify terpene profile variation across each THC:CBD chemotype, we computed the mean pairwise cosine distance in terpene profiles within each THC:CBD chemotype and used this as a measure of diversity. We conducted this analysis at the product level rather than sample level, as individual samples of the same product tend to be highly similar (see Methods). THC-dominant products displayed significantly higher levels of diversity than both balanced THC:CBD (Fig 5E; $P < 0.0001$, |d'| = 0.74) and CBD-dominant products (Fig 5E; $P < 0.0001$, |d'| = 0.89). This result held after performing a bootstrapping simulation, randomly sampling 245 products from each category to ensure equivalent sample sizes across groups. In particular, a higher proportion of CBD-dominant and balanced THC:CBD products displayed myrcene-dominant terpene profiles compared to THC-dominant samples (Fig 5F).

## Cluster analysis reveals distinct terpene chemotypes and poor validity of common commercial labels

Given the observed diversity of terpene profiles displayed by THC-dominant samples, we wanted to establish how this diversity is captured by the categorization system most used for commercial THC-dominant *Cannabis*. Commercial products are routinely labelled "Indica," "Hybrid," or "Sativa." Prevailing folk theories assert that "Indica" products provide sedating effects, "Sativa" energizing effects, and "Hybrids" intermediate effects [66]. If this were true, we would expect to see a reliable difference between the chemical composition of samples attached to each label. To test this, we devised an approach using silhouette analysis to quantify how well these industry labels capture the observed chemical diversity (see Methods). We compared this commercial labelling system to labelling the data with simplified chemical designations (each samples' dominant terpene), as well as an unbiased approach using k-means clustering.

Fig 6A displays THC-dominant samples plotted along the first two principal components, color-coded by their Indica/Hybrid/Sativa label. The samples are highly intermingled, with no obvious segregation of data points by commercial label. This is reflected in the corresponding silhouette plot, which displays a low mean silhouette score (Fig 6B). The majority of samples have a negative score, indicating that many samples with one label could be easily confused with samples of a different label in terms of terpene profile. In other words, it is likely that a sample with the label "Indica" will have an indistinguishable terpene composition as samples labelled "Sativa" or "Hybrid." By comparison, when samples are labelled by their dominant terpene, there is better visual separation of data points by their label (Fig 6C) and a higher mean silhouette score (Fig 6D). These results indicate that even a simplistic labeling system, in which THC-dominant samples are labelled by their dominant terpene, is better at discriminating samples than the industry-standard labelling system.

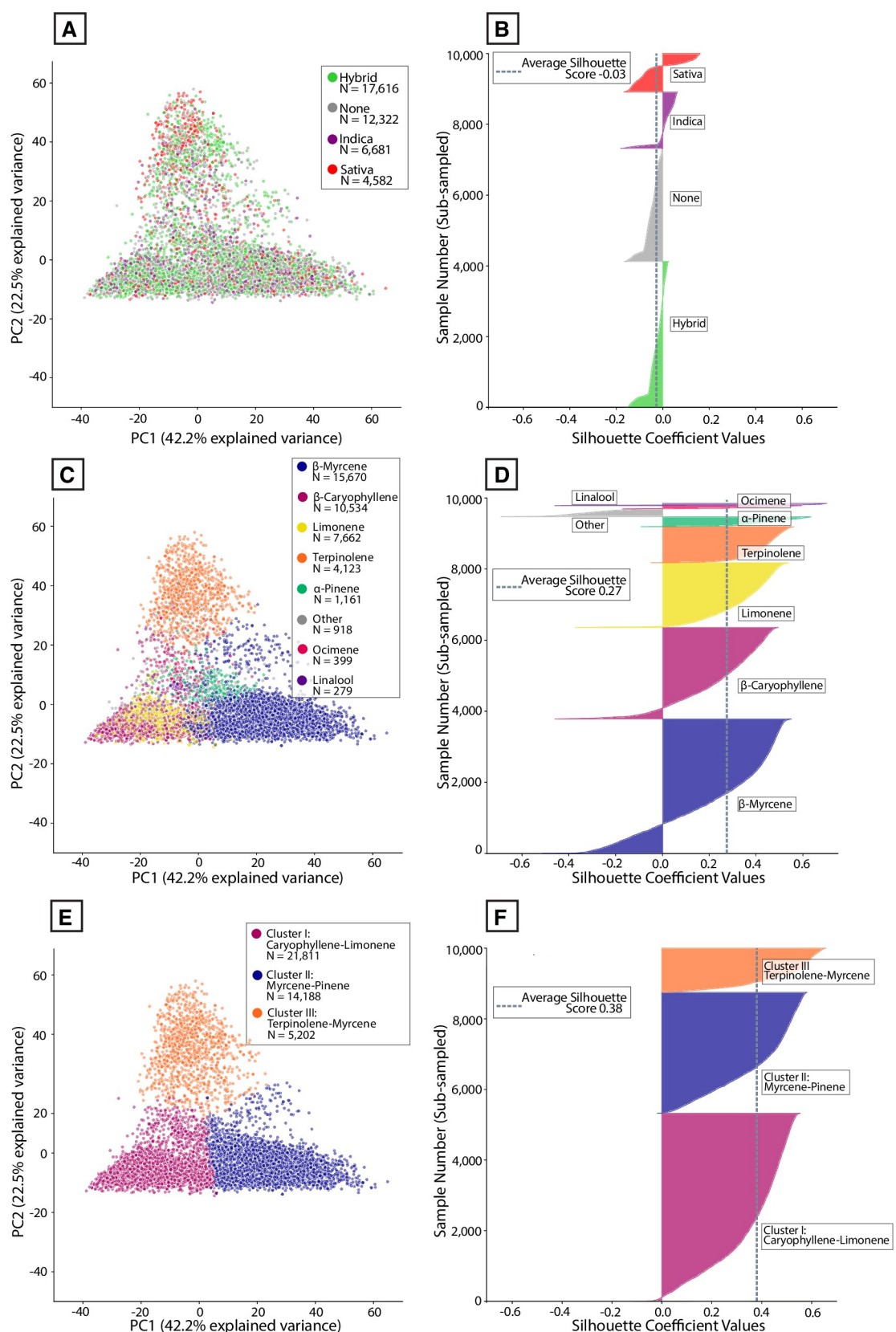

**Fig 6. Commercial "strain category" labels poorly align to patterns of phytochemistry. (A)** PCA scores for all THC-dominant samples plotted along PC1 and PC2, color-coded by Indica/Hybrid/Sativa label attached to each sample. **(B)** Silhouette coefficients for each sample with a given Indica/Hybrid/Sativa label. **(C)** PCA scores for all THC-dominant samples plotted along PC1 and PC2, color-coded by the dominant terpene of each sample. **(D)** Silhouette coefficients for each sample with a given dominant terpene. **(E)** PCA scores for all THC-dominant samples plotted along PC1 and PC2, color-coded by k-means cluster labels attached to each sample. **(F)** Silhouette coefficients for each sample with a given k-means cluster label. Each silhouette plot depicts a random subset of 10,000 samples from the full dataset ($n = 41,201$).

To segment samples in an unbiased fashion based on terpene profile, we applied the k-means clustering algorithm to define clusters of samples in the data. This approach allowed us to cluster the data using a standard method for determining what number of clusters fits this dataset well (Figs 6E, S5 and S6; see Methods). Three major clusters were defined. As expected, this algorithmic partitioning of the data is better at assigning points to distinct groups, especially compared to the Indica/Sativa labels. This is reflected in the higher mean silhouette score and very low proportion of samples with negative silhouette values (Fig 6F). This data can be clustered in different ways, such as defining additional sub-clusters within the clusters displayed here (S5 Fig). Ideally, this type of analysis would be further constrained by other data sources, such as sample genotypes and other classes of metabolites. For the purposes of this study, we focused on the three large clusters depicted in Fig 6 and conducted further analysis on their relationship to common commercial categories.

The distribution of silhouette scores across each of the three labelling systems allows us to compare the results depicted in Fig 6. Labelling data either by dominant terpene or by k-means cluster was significantly better at capturing the terpene diversity seen in THC-dominant samples compared to the commercial labels (Fig 7A; $P < 0.0001$, $|d'| = 3.49$, k-means vs. commercial labels). Regardless of the labelling system, samples are not evenly distributed among groups (Fig 7B). To further visualize the clusters defined in Fig 6E and 6F, we used Uniform Manifold Approximation and Projection (UMAP) to visualize the data (Fig 7B). UMAP is a dimensionality reduction technique like PCA but without linearity assumptions. The dimensions returned by UMAP lack the interpretability (e.g. factor loadings) associated with PCA but are superior at recovering latent clustered structure within high-dimensional data [67]. More of the individual data points are visible in this plot compared to the PCA plots shown in Fig 6.

Averaging the full cannabinoid and terpene profile of all products within each cluster allowed us to depict the average chemical composition of each cluster. We plotted mean terpene profiles as normalized polar plots together with the total THC, CBD, and CBG distributions of each cluster (Fig 7C–7F). In relative terms, a simplified description for the terpene profiles characterizing each cluster is: "high caryophyllene-limonene" (Cluster I), "high myrcene-pinene" (Cluster II), and "high terpinolene-myrcene" (Cluster III; Fig 4B–4D). Similar groups are seen across regional datasets (S6 Fig). We also observed that one cluster (Cluster III: "high terpinolene-myrcene") had somewhat higher total CBG levels compared to the other clusters (median CBG 0.98% vs 0.65%; $P < 0.0001$, $|d'| = 0.57$). This appeared to be due to a modest but significant correlation between total CBG and terpinolene levels ($r_s = 0.17$, $P < 0.0001$).

## Commercial "strain names" display differential levels of chemical consistency

The cannabis industry also uses colloquial "strain names" to label and market products. Distinct "strains" of THC-dominant *Cannabis* are purported to offer distinct psychoactive effects, such as "sleepy," "energizing," or "creative." While the commercial use of nomenclature is not

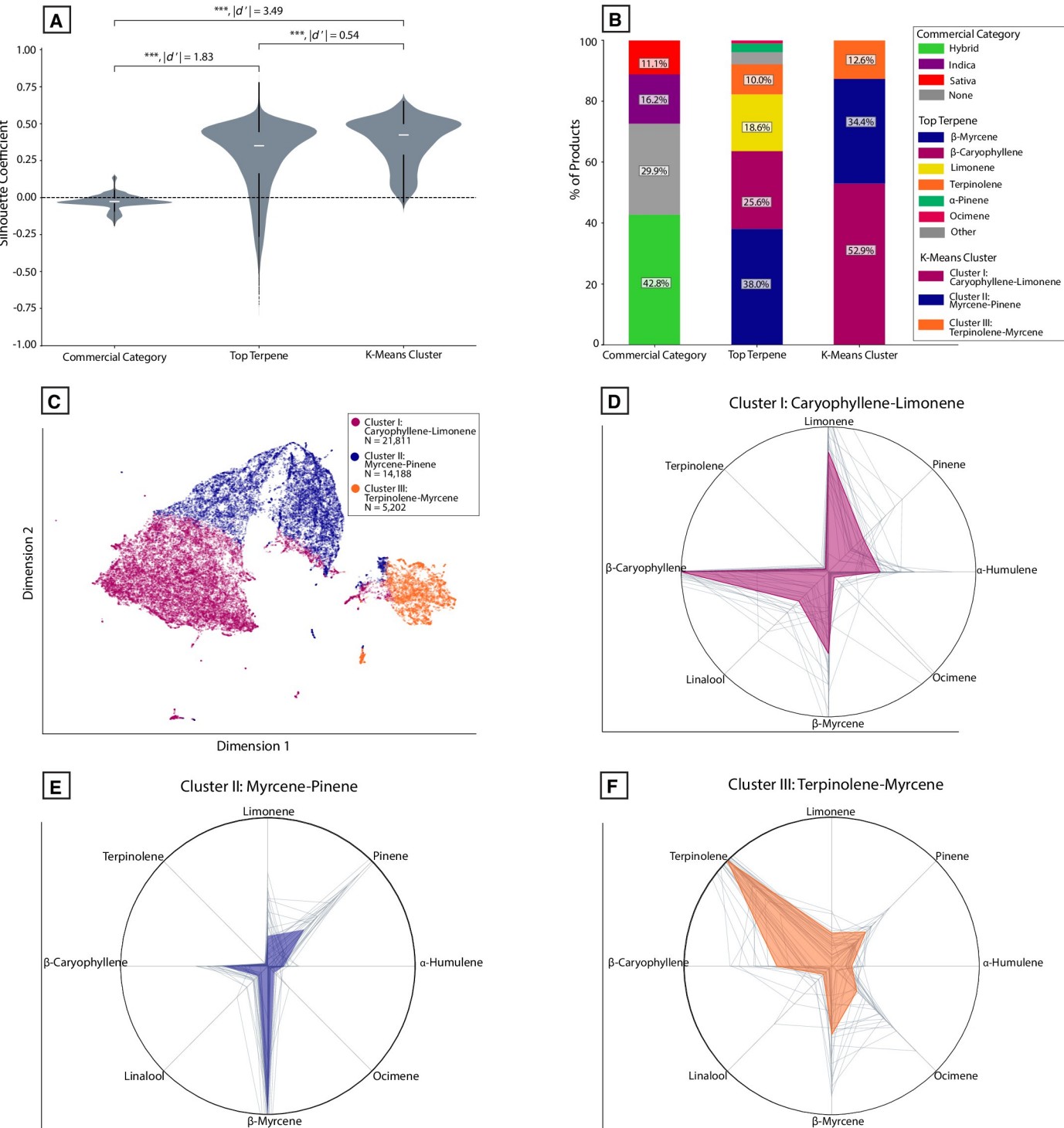

**Fig 7. Cluster analysis reveals distinct chemotypes of THC-dominant commercial *Cannabis* commonly present in US states. (A)** Violin plot showing the distribution of silhouette coefficients for each labelling method. ***$P < 0.0001$, Welch's t-test and Cohen's d'. Absolute effect sizes are given as Cohen's d' values. ***$P < 0.0001$, **$P < 0.001$; *$P < 0.01$ **(B)** Stacked bar chart showing the percent of samples falling within each group for each labelling system. **(C)** UMAP embedding in two dimensions showing samples classified into each k-means cluster. **(D)** Polar plot showing the mean, normalized levels of eight of the most abundant terpenes observed for Cluster I (high caryophyllene-limonene) products. **(E)** Similar polar plot for Cluster II (high myrcene-pinene) products. **(F)** Similarly polar plot for Cluster III (high terpinolene-myrcene) products. Gray lines represent the top 25 products from each cluster with the most samples per product.

accepted by the scientific community, it is conceivable that distinct chemovars of THC-dominant *Cannabis* could cause different psychoactive effects, on average. In principle, if commercial "strain names" are indicative of different psychoactive effects in a discernible way, then these labels should reliably map to distinct chemotypes. Alternatively, because there are few regulatory constraints on the nomenclature of commercial *Cannabis*, it is possible that *Cannabis* cultivators label their products in arbitrary or inconsistent ways. If this were true, we would not expect to see "strain names" consistently map to specific chemotypes above chance levels.

To quantify chemical consistency among THC-dominant products, we compared each product's chemical composition in terms of the 14 major terpenes depicted in Figs 3 and 4. We did this for all "strain names" where the underlying data was attached to at least five cultivator IDs each having five or more samples with that particular name. To validate whether the names attached to more testing data are representative of those encountered by consumers, we plotted the number of products attached to each name vs. consumer popularity, measured in terms of unique online pageviews to a widely used consumer *Cannabis* database (Fig 8A). We observed a strong positive correlation ($r_s$ = 0.59, $P < 0.0001$), indicating that the names in our analysis are representative of the names encountered by consumers in commercial settings.

As a measure of consistency, we computed the pairwise cosine similarity of all products attached to each "strain name" and visualized this in a similarity matrix (Fig 8B, ten most abundant strain names shown). Next, we quantified the average pairwise similarity of all products sharing a common strain name. For each strain name, we plotted the distribution of product similarity scores, sorted from highest to lowest mean similarity, for the 41 "strain names" used in this analysis (Fig 8C). We compared these values to the average similarity score computed after randomly shuffling names across all cultivator IDs (Fig 8C, dashed line). This allowed us to model the situation where each cultivator has arbitrarily labelled their product with a given name. The mean between-product similarity was significantly higher compared to the shuffled dataset for the majority of names (Fig 8C, $P < 0.0001$, $|d'| = 1.44$). For some names, product similarity did not significantly differ from the shuffled distribution or was even below this, and there was a large amount of variability in mean consistency scores across all "strain names." To illustrate this variability further, we overlaid the individual profiles of all products with a given name, separately for two strain names: one with a relatively high level of between-product similarity ("Purple Punch") and one with a low level ("Tangie"; Fig 8D).

To assess between-product similarity in terms of the major clusters defined previously, we applied the same clustering approach from Figs 6 and 7 to the product averages analyzed in Fig 8. These data were visualized in a UMAP embedding, with all products attached to the two example "strain names" (Fig 8E). This illustrates how a relatively consistent (Purple Punch) vs. inconsistent (Tangie) name maps to this space. 96% of product averages attached to "Purple Punch" fall within Cluster I (high caryophyllene-limonene), while only 62.5% of product averages for "Tangie" fall into a single cluster.

## Some commercial labels are overrepresented in specific chemically defined clusters

To further understand whether any "strain names" were overrepresented in our algorithmically defined clusters, as appeared true for Purple Punch (Fig 8E), we calculated the proportion of all products with a given name that belonged to each cluster. For each name displayed in Fig 8C, we calculated that proportion for whichever cluster contained the highest count of products with that name. For example, 96% of products attached to the name "Purple Punch" were found in Cluster I, much higher than the 61.8% expected if product strain names are randomly shuffled ($P < 0.0001$, $|d'| = 2.47$). We plotted this proportion for the 18 most overrepresented

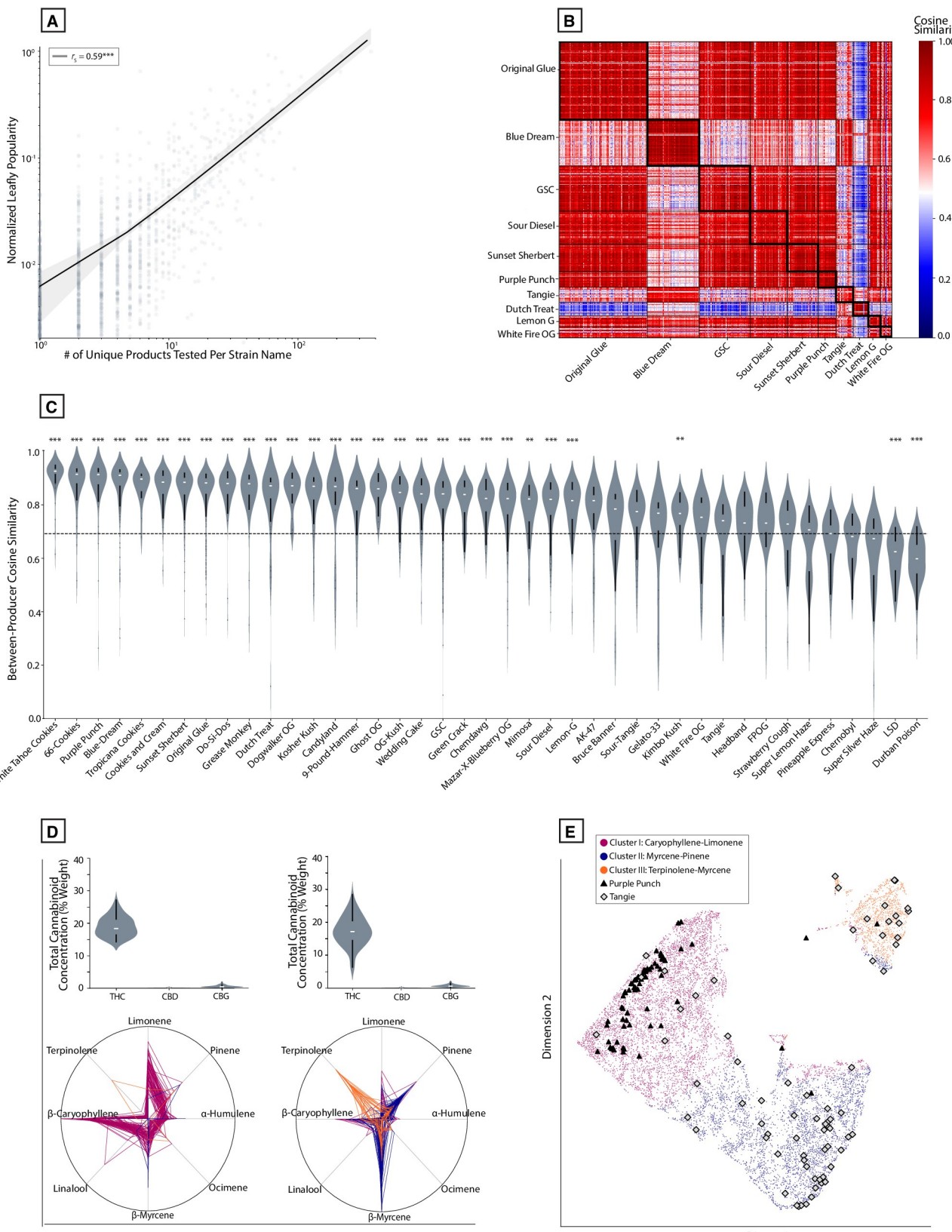

**Fig 8. Commercial "strain names" are associated with variable levels of chemical consistency across *Cannabis* products. (A)** Scatterplot of the number of products tested vs. normalized popularity for all product-level data attached to cultivator-given strain names ($\log_{10}$ scale). $r_s = 0.59$, ***$P < 0.0001$ **(B)** Similarity matrix depicting pairwise cosine similarities between all product-level data attached to the ten most common strain names by abundance. **(C)** Violin plot depicting the distribution of cosine similarity scores between products attached to the same strain name. Dashed line represents the average similarity level after randomly shuffling strain names. **$P < 0.001$, ***$P < 0.0001$, Welch's t-test. **(D)** Violin plots representing total cannabinoid distributions and polar plots representing terpene profiles for all products attached to the strain names "Purple Punch" (left) and "Tangie" (right); **(E)** UMAP embedding showing where each of the product samples for Purple Punch and Tangie from panel D show up in this representation.

names, grouped by their primary cluster and compared these to the average cluster frequency expected from shuffling names across products (Fig 9D). For each cluster, there are "strain names" that are highly overrepresented. For example, 100% of "Dogwalker OG" products are found within Cluster I ("high caryophyllene-limonene"; $P < 0.0001$, $|d'| = 1110.4$), 88.5% of "Blue Dream" products are found within Cluster II ("high myrcene-pinene"; $P < 0.0001$, $|d'| = 1.2$), and 85.9% of "Dutch Treat" products are found within Cluster III ("high terpinolene"; $P < 0.0001$, $|d'| = 1.0$).

As in Fig 8E, we plotted the single most overrepresented "strain name" associated with each cluster in a UMAP embedding of all the product-level data (Fig 9C). These names represent those that are the most consistently associated with a given chemotype. Notably, even these "strain names" are not perfectly associated with a single chemotype. Products attached to each name also display variability within each cluster. This indicates that even the names with the highest levels of consistency across products may display a non-trivial level of variation. An interactive 3-D version of this product-level UMAP graph (including high-CBD products) is also included (see Methods).

In doing this analysis, we noticed that one cluster (Cluster III, characterized by high terpinolene levels) contained a paucity of products labelled as "Indica." To understand whether any of the Indica/Hybrid/Sativa industry labels were over- or under-represented within any of these clusters, we performed a similar analysis for commercial categories as we did for "strain names": for each of the three clusters, we calculated the proportion of products attached to Indica/Hybrid/Sativa labels. For each of these, we compared it to the population frequency of each category. For Cluster I and Cluster II, the frequency of products attached to Indica/Hybrid/Sativa labels did not significantly differ from those observed in the full set of products. In contrast, Cluster III (high terpinolene) did show a significant difference, with approximately twice as many Sativa-labelled products and half as many Indica-labelled products as expected from the full population (Fig 9B; $X^2 = 22.2$, $P < 0.0001$, Chi-squared test). This over-representation of Sativa-labelled products can also be seen in the UMAP embedding (Fig 9A), which displays product-level data color-coded by Indica/Hybrid/Sativa label.

## Discussion

To our knowledge, this study represents the largest quantitative chemical mapping of commercial dispensary-grade *Cannabis* flower samples to date. It builds on a literature examining the chemotaxonomy of *Cannabis*-derived samples taken from individual regions of the US [35, 46, 48], Canada [50], and Europe [43, 49], as well as classic studies of the chemotaxonomy of non-commercial *Cannabis* [33, 45]. We mapped and analyzed the cannabinoid and terpene diversity of almost 90,000 samples from six US states and found distinct chemotypes of *Cannabis* that are reliably present across regions.

Because *Cannabis* remains federally illegal in the US, the laboratory-derived data from each state represent distinct pools of *Cannabis* found within those states. Even with clones, environmental factors such as variation in growing conditions and preparation procedures can cause

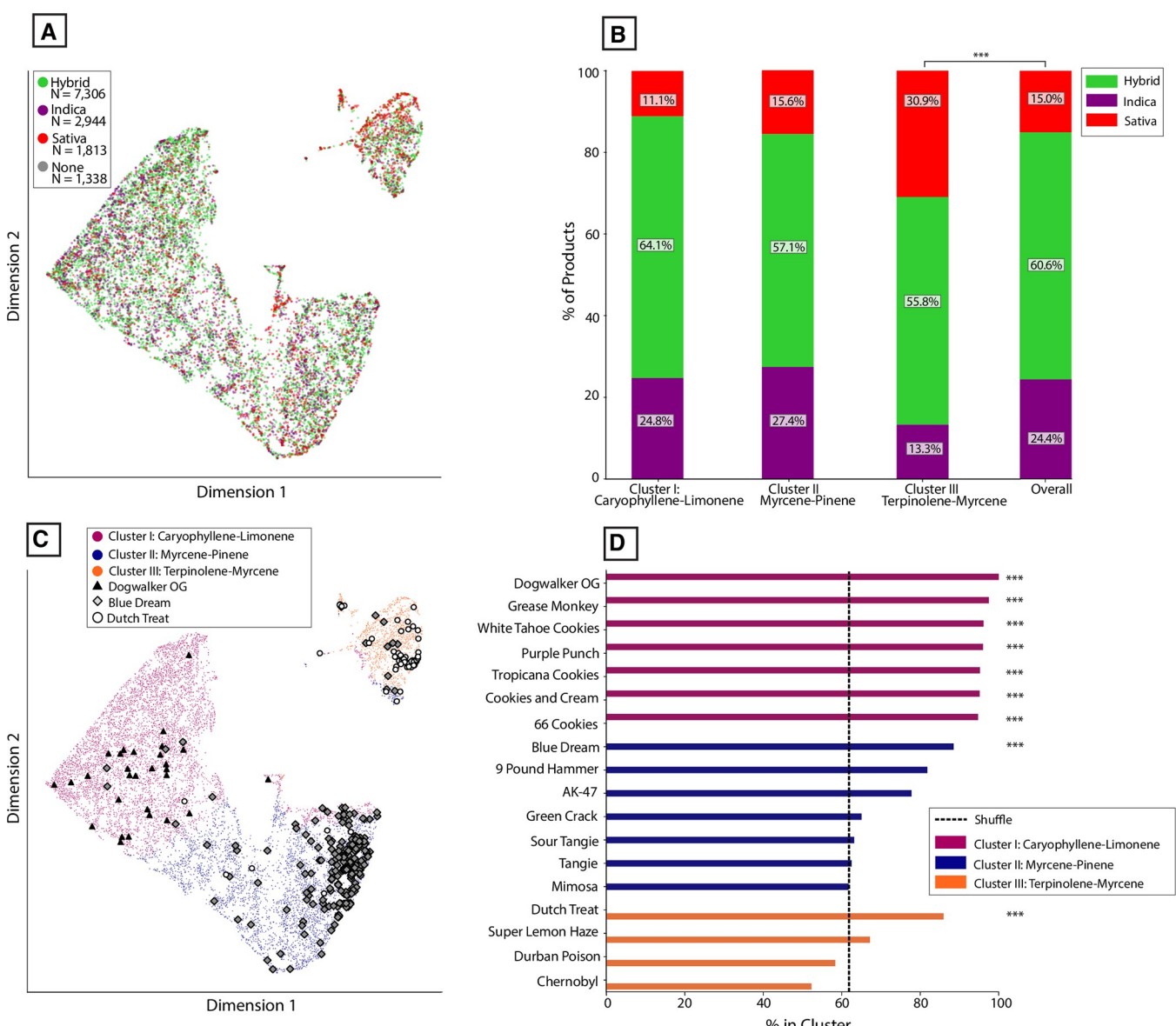

**Fig 9. Some commercial *Cannabis* labels are overrepresented for specific chemotypes. (A)** UMAP embedding of product-level data as in Fig 8E, color-coded by Indica/Hybrid/Sativa label. **(B)** Stacked bar chart showing the proportion of products labelled as Indica, Hybrid, or Sativa within each k-means cluster, compared to the overall distribution. ***$P < 0.0001$, Chi-squared test. **(C)** UMAP embedding of product-level data as in Fig 8D, color-coded by k-means cluster label, showing where all products attached to either "Blue Dream" or "Dutch Treat" are found. **(D)** Bar charts showing the percent of products attached to each strain name that are found in each k-means cluster, color-coded by its most prominent cluster. Dashed line represents expected percent after randomly shuffling strain names. ***$P < 0.0001$, Welch's t-test.

differences in morphology and chemotype expressions that are measured by testing labs [68]. Importantly, the measurements themselves are made by different labs, using methodologies that may not be standardized (See Methods, Data Collection). Thus, we cannot discern whether regional differences, such as finer-scale differences in terpene profile patterns (S6 Fig), reflect true regional sample variation vs. measurement differences between labs. Nonetheless, we observed similar patterns across regions. In all states, the sample population is comprised mostly of THC-dominant samples, each with a similar distribution of major terpenes (S2 and S6 Figs) and displaying the terpene-terpene correlations expected based on the

constraints of terpene biosynthesis [63, 64, 69], as has been observed elsewhere [50, 65]. The pooled dataset also displays features seen in sample populations from US states not represented here [35]. Collectively, these results suggest that, while some regional variation may exist, the major patterns of cannabinoid and terpenes profiles are similar throughout the US.

We used cluster analysis to define at least three major chemotypes of THC-dominant *Cannabis* prevalent in the US (Figs 6, 7 and S5). In simplified terms, samples from each cluster tend to be characterized by relatively high levels of β-caryophyllene and limonene (Cluster I), myrcene and pinene (Cluster II), or terpinolene and myrcene (Cluster III). Samples across these clusters display similar total THC distributions, while Cluster III is associated with modestly higher CBG levels (Fig 7). The chemotype landscape of commercial *Cannabis* is highly uneven, with 96.5% of samples classified as THC-dominant and 87.4% of these samples belonging to either the Cluster I (high caryophyllene-limonene) or Cluster II (high myrcene-pinene). Breeding new *Cannabis* chemotypes not represented in the current commercial landscape will be a key area of future innovation. In agreement with other studies [36, 42, 70], or results indicate that terpene composition is an effective way of clustering varieties.

We observed that the diversity of cannabinoid profiles displayed by commercial *Cannabis* in the US is explained almost entirely by variation in total THC, CBD, and CBG content, with most variation coming from THC content (Fig 1). Similar to classic work on non-commercial *Cannabis* [33], our results show distinct THC:CBD chemotypes: THC-dominant ("Type I"), balanced THC:CBD ("Type II"), and CBD-dominant ("Type III"). These likely arise from distinct genotypes. The genes giving rise to the cannabinoid synthases responsible for producing the major cannabinoid acids are highly similar [20, 71, 72]. Copy number variation [20, 73] or allelic variation [19] in the genes encoding these enzymes may explain the observed variation in cannabinoid ratios. Interesting areas of future study will be to correlate chemotype and genotype directly and determine why other cannabinoids have such low abundance in commercial *Cannabis*. For example, there are numerous CBC-related genes [72] but we observe very low levels of CBC (Figs 1 and 2), supporting previous claims that CBCA synthase may not be selective for CBC production [46].

The observed variation in terpene profiles is also likely related to underlying genotypic variation. While environmental and developmental modulation of terpene profiles is possible [74], the fact that we observe a similar set of major profiles across US states (S6 Fig) suggests that these profiles have a strong genetic component. *Cannabis* terpenes are synthesized from enzymes encoded by multiple genes [63–65, 69]. The robust correlation patterns we observed among many of the most abundant *Cannabis* terpenes likely arise from variation in biosynthetic enzymes. The underlying genetic networks regulating these biochemical pathways are complex [63–65, 69, 75] and more research may be needed to inform efficient breeding programs to generate novel chemotypes.

Despite the chemotypic diversity we observed for THC-dominant *Cannabis*, this likely represents a fraction of the diversity the plant can express. For example, although one of the clusters we defined is characterized by especially high myrcene levels, each of the three clusters contain samples where myrcene is more abundant than most other terpenes. This pattern is stronger for CBD-dominant and balanced THC:CBD chemotypes, where the majority of samples are myrcene-dominant. This may reflect a historical genetic bottleneck, whereby most *Cannabis* grown in the US is descended from a subset of worldwide lineages [66]. The relative lack of diversity among high-CBD cultivars is likely due to the historical focus on breeding high potency THC-dominant *Cannabis* in the US. In principle, there is no biological limitation preventing the breeding of high-CBD cultivars with similar terpene diversity to what is seen in THC-dominant cultivars. Many of the genes encoding the synthetic enzymes for terpene production are located on different chromosomes from those involved in cannabinoid acid

synthesis [69] or are found far apart from each other in the same genomic region [65], and therefore could be assorted through recombination. These two aspects of chemical phenotype may therefore be independently inherited as seen for other phenotypic traits [76].

While not observed in this commercial dataset, chemovars that predominate in other cannabinoids, such as CBG, have been bred and may offer distinct psychoactive or medicinal effects compared with the high-THC chemovars that predominate commercially [10]. There were few samples that contained an abundance of minor cannabinoids, suggesting that commercial *Cannabis* in the US is much more homogenous than it could be. An exciting area for academic research and product innovation lies in the breeding of new varieties with higher levels of other cannabinoids. For example, cannabinoids like THCV have interesting pharmacological properties suggesting they may be dose-dependently psychoactive [61], with potential medicinal benefits [77]. Chemotypes expressing distinct ratios of minor cannabinoids and terpenes, with and without significant THC levels, will likely elicit effects of interest to consumers and clinical researchers. Our results are consistent with the notion that the full chemotype landscape of *Cannabis* has yet to be filled in (Fig 10).

In addition to mapping the chemical landscape of commercial *Cannabis* in the US, we also quantified how well commonly used industry labels align with the chemical composition of samples. In general, we found that industry labels are poorly or inconsistently aligned with the underlying chemistry. In particular, the Indica/Hybrid/Sativa nomenclature does not reliably distinguish samples based on their chemical content, making it highly unlikely that this widely used commercial labeling system is a reliable indicator of systematically different effects. This is in agreement with other studies [71]. Marketing emphasizing Indica-labelled products as sedating and Sativa-labelled products as energizing is specious given our analysis of the underlying chemistry.

We also examined the popular "strain names" commonly attached to products, which are used commercially to reference cultivars purported to offer distinct effects. We quantified the terpene profile consistency of THC-dominant products sharing the same strain name across different producers. We modeled the situation where strain names are randomly applied to products, finding that many strain names are more consistent from product-to-product, on average, than would be expected by chance. However, we also observed a wide range of consistencies for all strain names, suggesting that some are more homogeneous than others [78], perhaps because these names are more often attached to cultivars that are clonally propagated. These results indicate that while strain names may be a better marker of product chemistry than the Indica/Sativa/Hybrid category labels, they are far from ideal (Fig 8).

While commercial labels tended to have poor validity overall, we found evidence that certain names and categories were statistically overrepresented within specific chemically defined clusters. Cluster III samples (high terpinolene-myrcene) displayed an over-representation of Sativa-labelled products. While certain "strain names" were overrepresented in Clusters I and II, neither of these Clusters displayed an over-representation of Indica or Sativa labels. Although the origins of this pattern are unclear, one hypothesis is that it echoes patterns of phytochemistry that may have been more distinctive prior to the long history of *Cannabis* hybridization in the US. It is conceivable, for example, that certain cultivars commonly associated with "Sativa" lineages may have historically displayed a chemotype reliably distinct from those in other lineages. Over time, hybridization and a lack of standardized naming conventions may have decorrelated chemotaxonomic markers from the linguistic labels used by cultivators. Thoroughly tracing which chemotypes tend to map to different lineages will require datasets that combine both genotype and chemotype data for modern commercial cultivars and, ideally, the landrace cultivars from which they descended [2].

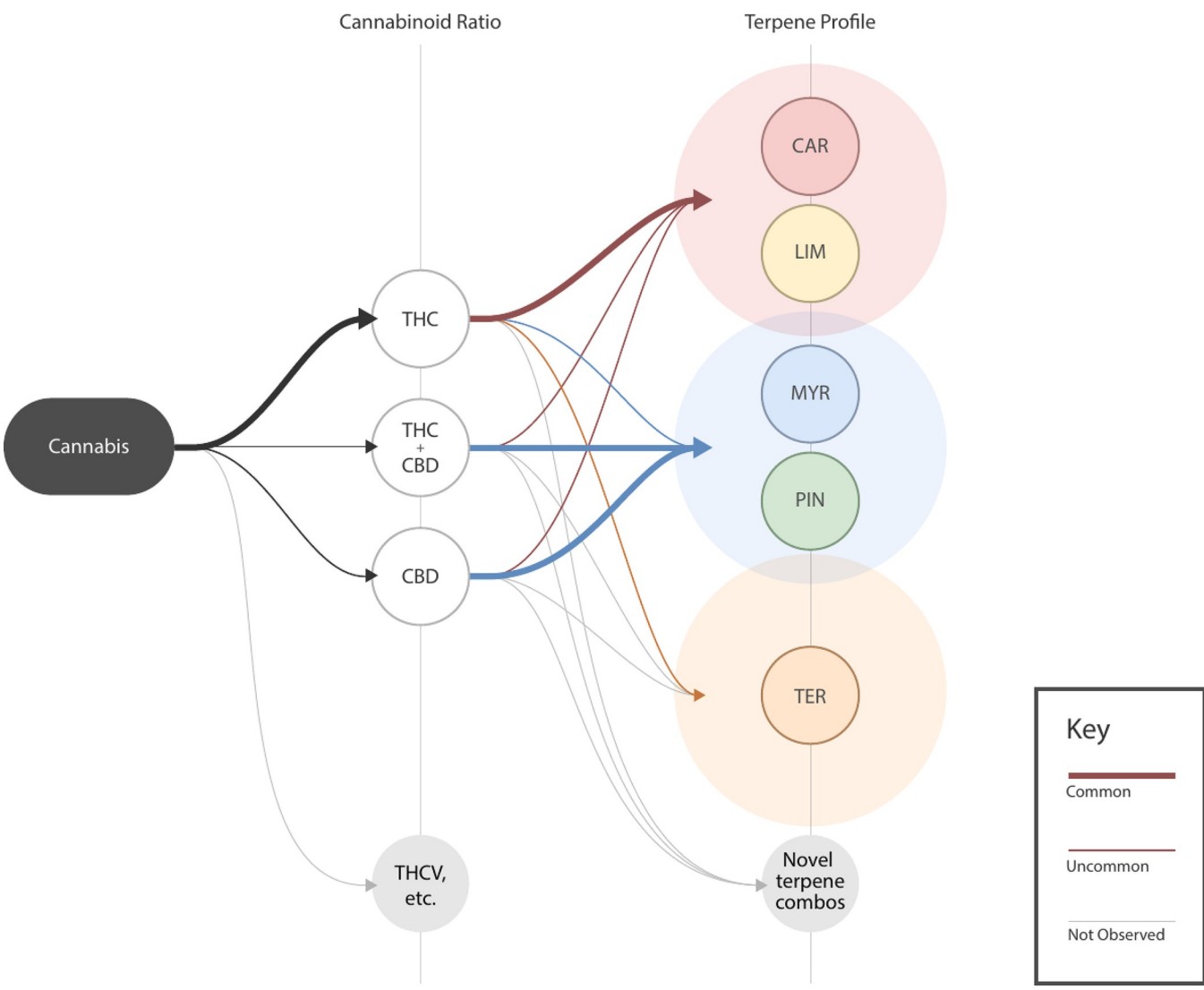

**Fig 10. Potential scheme for classifying commercial *Cannabis* based on cannabinoid and terpene profiles.** Flow chart showing a potential classification framework for commercial *Cannabis*. Level 1 represents cannabinoid ratios and displays the three common THC:CBD chemotypes as well as novel cannabinoids that could be bred. Level 2 represents terpene profiles and displays the three clusters we identified as well as other terpene combinations which could come to exist. Terpene clusters overlap slightly to illustrate that terpenes in each cluster are not mutually exclusive. Grey lines demonstrate a chemotype that may be possible (e.g., CBD-dominant and terpinolene-dominant) but has not yet been observed.

Medical *Cannabis* has been described as a "pharmacological treasure trove" [79] due to the diversity of pharmacologically active compounds it harbors. *Cannabis*-derived formulations and specific cannabinoids (namely THC and CBD) have demonstrated efficacy for conditions ranging from chronic pain [80] to childhood epilepsy [81]. Medical *Cannabis* patients report an even wider array of conditions they believe *Cannabis* is efficacious for, including mental health outcomes [82]. It has also been hypothesized that distinct chemotypes of *Cannabis*, each with different ratios of cannabinoids and terpenes, may offer distinct medical benefits and psychoactive effects [55, 83]. This hypothesized "entourage effect" has been difficult to confirm experimentally due to onerous regulations that make it challenging to execute *in vivo* studies with controlled administration of the myriad compounds found in *Cannabis*.

The results of this study can serve as a guide for future research, including *in vitro* assays, animal studies, and human trials. Studies seeking to falsify claims about the psychoactive and medical effects of different *Cannabis* types should test chemical ratios that match those found commercially. If it is true that different chemotypes of THC-dominant *Cannabis* reliably produce distinct psychoactive or medicinal effects, then a sensible starting point is to design studies comparing the effects of common, distinctive commercial chemotypes, such as those described by our cluster analysis (Figs 6 and 7). Likewise, if there is any modulatory effect of specific cannabinoids or terpenes on the effects of THC, then this should be tested using formulations designed to match the ratios that people choose to consume under 'ecological' conditions.

While the present study represents the largest chemotaxonomic analysis of commercial *Cannabis* to-date, there are important caveats. One is that the dataset we analyzed was an aggregation of lab data from different states. We had no access to the genotype or the growing conditions for any of these samples and important outstanding questions remain for how these factors relate to chemotype in *Cannabis*. It is also possible one or more compounds that were not consistently measured in each region is an important chemotaxonomic marker. State-level markets have different regulations which may influence the expertise of commercial cultivators or the choice and development of *Cannabis* products. Finally, this dataset did not include the variation found in hemp. An exciting area of future research will be to investigate these questions using datasets that combine sample-level features about genotype, chemotype, and environmental conditions.

Our results also have regulatory implications. For example, we observed a robust correlation between total THC and total CBD levels for CBD-dominant *Cannabis* samples. Because the legal definition of hemp in the US is based on a discrete threshold of total THC levels, many CBD-dominant samples would not be legally classified as hemp within the US, despite such samples being characterized by low THC:CBD ratios distinct from those seen in high-THC samples (Figs 1 and 2).

Legal THC-dominant *Cannabis* products are marketed to consumers as if there are clear-cut associations between a product's label and its psychoactive effects. This is deceptive, as there is currently no clear scientific evidence for these claims and our results show that these labels have a tenuous relationship to the underlying chemistry. In contrast to other widely used but federally regulated plants (e.g., corn and other crops regulated by the Federal Seed Act), there are no enforced rules for the naming of *Cannabis* varieties. This stems from the fact that *Cannabis* is not federally legal in the US, which prevents an overarching, enforceable naming standard from emerging. Consequently, legacy classification systems inherited from the illicit market have persisted with unwarranted trust in the provenance and predictability of products' effects.

We have shown that in the US, multiple, distinct chemotypes of commercial *Cannabis* are reliably present across regions. Due to the chemical complexity of these products, which may contain dozens of pharmacologically active compounds with potentially psychoactive or medicinal effects, we believe it is in the public interest to devise a classification system and naming conventions that reflect the true chemotaxonomic diversity of this plant. The general approach we have used in this study can serve as a basic guide for cannabis product segmentation and classification rooted in product chemistry. Consumer-facing labelling systems should be grounded in such an approach so that consumers can be guided to products with reliably different sensory and psychoactive attributes.

## Materials and methods

### Data collection

The data analyzed in this paper was shared by Leafly, a technology company in the legal cannabis industry, as part of a data sharing program where university-affiliated researchers can

access data for research purposes with the intent to publish results in peer-reviewed scientific journals. This dataset included laboratory testing data (cannabinoid and terpene profiles; see below) as well as metrics related to consumer behavior and preferences, including: normalized values of the number of unique views to each of the web pages within its online, consumer-facing strain database; consumer ratings and common categorical designations associated with commercial strain names (Indica, Hybrid, or Sativa); crowd-sourced metrics related to the perceived flavors and effects of associated with popular strain names, derived from online consumer reviews.

For the purposes of this study, we focused mainly on analyzing the laboratory testing data and its relationship with popular commercial labelling systems (i.e. strain names and Indica/ Hybrid/Sativa designations). There was a single lab dataset composed of *Cannabis* flower samples that had been tested for cannabinoid, or for both cannabinoid and terpene content. "Flower" refers to products which are the dried mature female flower of the *Cannabis* plant. The source material for these measurements is composed of this material, which are samples grown by commercial *Cannabis* producers, tested to be compliant with state laws, and intended for distribution to licensed retailers (often referred to as "dispensaries") where they can be purchased by consumers. This is the most widely consumed product type in the legal cannabis industry, as well as the type most closely resembling the intact plant. Products derived from flower, such as "joints", "edibles", or "concentrates", were not included in analysis.

The counts for flower samples from each lab before dataset cleaning are found in Table 1. Raw cannabinoid acid, cannabinoid, and terpene measurements had been converted to common units (% weight) together with additional information for each sample: anonymized producer ID, test date, and the producer-given sample name.

Each lab supplied cannabinoid and cannabinoid acid measurements in units of percent of sample weight (% weight), parts-per-million (ppm), or milligrams per gram (mg/g). These measurements were converted to % weight if not already in that unit. These measurements were obtained by each lab using High-performance liquid chromatography (HPLC) as the chromatographic method for separating individual cannabinoids and cannabinoid acids contained within samples. Each lab used either diode array detection (HPLC-DAD) or ultraviolet-visible spectroscopy (HPLC-UV) as detection methods (see Table 1 above). Terpene measurements were obtained using Gas Chromatography with a flame ionization detector (GC-FID) or mass spectrometry (GC-MS). The number of raw samples with cannabinoid and terpene measurements, unique cultivators, and unique cultivar names contained in the datasets from each testing lab are shown in Table 2 below.

For each sample, the popular "strain names" associated with each sample were included, together with the popular industry category ("Indica," "Hybrid," or "Sativa") associated with each name. These labels were matched to the producer-given cultivar names of each flower sample (e.g. "blue-dream"), wherever such a match was found, using a similar string-matching

**Table 1. Chemical measurement method by testing lab.**

| Testing Lab | State | Cannabinoid Method (units) | Terpene Method (units) |
|---|---|---|---|
| CannTest | AK | HPLC-DAD (% weight) | GC-FID (% weight) |
| ChemHistory | OR | HPLC-DAD (% weight) | GC-FID (% weight) |
| Confidence Analytics | WA | HPLC-UV (% weight) | GC-FID (ppm) |
| Modern Canna Science | FL | HPLC-UV (% weight) | GC-FID (% weight) |
| PSI Labs | MI | HPLC-DAD (% weight) | GC-MS (ppm) |
| SC Labs | CA | HPLC-DAD (mg/g) | GC-FID (mg/g) |

**Table 2. Sample, cultivator, and cultivar breakdown by testing lab, before data cleaning.**

| Testing Lab | State | # of samples with cannabinoid measurements | # of samples with cannabinoid and terpene measurements | # of unique cultivators | # of unique cultivar names |
|---|---|---|---|---|---|
| CannTest | AK | 6,255 | 6,202 | 296 | 836 |
| ChemHistory | OR | 13,546 | 13,544 | 591 | 1,539 |
| Confidence Analytics | WA | 53,233 | 11,138 | 847 | 1,800 |
| Modern Canna Science | FL | 1,620 | 886 | 5 | 121 |
| PSI Labs | MI | 7,243 | 7,241 | 544 | 749 |
| SC Labs | CA | 8,119 | 8,119 | 1,060 | 1,218 |
| **Total** | | 90,016 | 47,130 | 3,343 | 3,090 |

algorithm as described in Jikomes & Zoorob (2018), supplemented with a human expert-supplied dictionary used to standardize names with common variations (e.g. "SLH" = "super-lemon-haze," "GDP" = "granddaddy-purple," and so on). In total, 81.5% of samples were attached to popular strain names and 73.4% additionally attached to a Indica/Hybrid/Sativa label, with the remainder labelled as "Unknown." There were 3,087 unique cultivar names represented in the dataset before cleaning.

Laboratory testing data originated from cannabis testing labs across the US. Each lab consented to allowing researchers to analyze its data for academic research purposes. Each laboratory dataset consisted of the complete set of cannabinoid and terpene compounds measured by each lab within a given time period between December 2013 and January 2021. In addition to the general summary of the lab methods given above for both cannabinoid and terpene measurement, we have included more detailed summaries of their testing methods on Github (https://github.com/cjsmith015/phytochemical-diversity-cannabis). Each lab consented to allowing us to share the dataset we used for analysis in this study (see below).

## Technologies used

All data cleaning and analysis for this paper was performed using the Python programming language (Python Software Foundation, https://www.python.org) and utilized the following libraries: NumPy, pandas, SciPy, and scikit-learn. All data visualizations were made using the Python libraries Seaborn and Matplotlib. Coding for visualizations and analysis can be found on the study's Github repository (https://github.com/cjsmith015/phytochemical-diversity-cannabis/).

## Data processing

**Raw data filtering and outlier removal.** The testing lab dataset, consisting of rows of cannabinoid and terpene measurements, was cleaned and processed using custom code in Python. A small number of duplicate rows were removed from the dataset ($n = 11$). We also removed any samples with biologically implausible values (i.e. very high or low) for dried *Cannabis*, which likely represent rare measurement anomalies or come from samples which do not truly represent dried *Cannabis* flower (e.g. "shake" or other plant material different from the dried female inflorescence). We used the following, conservative criteria: any single cannabinoid measured at over 40% (percent weight; $n = 80$), or samples which had summed total cannabinoid measurements over 50% ($n = 2$); samples which had null or 0.0 measurements for both total THC and total CBD ($n = 591$). The total number of samples dropped from the dataset was 684, or 0.75% of the raw dataset. The final number of samples was 89,923.

**Table 3. Sample, cultivator, and cultivar breakdown by testing lab, after data cleaning.**

| Testing Lab | State | # of samples with cannabinoid measurements | # of samples with cannabinoid and terpene measurements | # of unique cultivators | # of unique cultivar names |
|---|---|---|---|---|---|
| CannTest | AK | 6,253 | 6,173 | 293 | 834 |
| ChemHistory | OR | 13,508 | 11,720 | 589 | 1,538 |
| Confidence Analytics | WA | 53,190 | 11,070 | 831 | 1,794 |
| Modern Canna Science | FL | 1,620 | 695 | 5 | 121 |
| PSI Labs | MI | 7,240 | 5,268 | 543 | 748 |
| SC Labs | CA | 8,112 | 7,917 | 1,058 | 1,218 |
| **Total** | | 89,923 | 42,843 | 3,319 | 3,087 |

Terpene data was also removed for samples which had a terpene measurement variance less than 0.001 (n = 2,048), samples which had any single terpene measurement over 5% (n = 8), or for samples which had over 10 measurements equaling zero among the 14 most common terpenes (n = 2,178). The total number of samples which had terpene data removed was 4,234, or 9% of samples having any terpene data. The final number of samples with terpene data was 42,843, or 47.6% of the final dataset. The reason that many laboratory testing samples contain only cannabinoid measurements is that terpene levels are generally not legally required to be measured. Nonetheless, we were still left with 42,843 samples with terpene measurements attached, which to our knowledge is the largest such dataset of commercial Cannabis analyzed to date. The counts for flower samples, unique cultivators, and unique cultivar names from each lab after dataset cleaning are found in Table 3.

The final, cleaned dataset is available on the study's Github repository (https://github.com/cjsmith015/phytochemical-diversity-cannabis/) and contains all data necessary to run the analysis for Figs 1–9 as well as S2 and S4–S6 Figs. S1 and S3 Figs contain information specific to labs. The data underlying the results for S1 and S3 Figs are available upon request, with consent from each individual lab.

**Total cannabinoid levels.** Total cannabinoid levels were calculated from the raw cannabinoid and cannabinoid acid values attached to each flower sample. This widely used convention calculates the total levels of a cannabinoid found in a *Cannabis* product assuming complete decarboxylation of a cannabinoid acid to its corresponding cannabinoid. For total THC, the formula is:

$$\text{Total THC} = (0.877 * \text{THCA}) + \text{THC}$$

0.877 is a scaling factor which accounts for the difference in molecular weight between raw cannabinoid and cannabinoid acid values for THC, CBD, CBG, CBC, CBN, CBT, and delta-8 THC. The equivalent formula, with the scaling factor of 0.8668, was used to calculate total cannabinoid levels for THCV and CBDV.

**THC:CBD chemotypes.** Following past work [33, 34], we classified all flower samples as THC-dominant, CBD-dominant, or Balanced THC:CBD based on the THC:CBD ratio of the sample. THC-dominant samples are those with a 5:1 THC:CBD or higher, CBD-dominant samples are those with a 1:5 THC:CBD or lower, and Balanced THC:CBD are in between.

## Data analysis

**Cannabinoid and terpene analysis.** Given that cannabis testing is not standardized nationally, each lab had a unique set of cannabinoids and terpenes that they measured. Because

of this, we established a list of compounds common across every lab and used these in our main analyses. These common cannabinoids were: Δ-9-tetrahydrocannabinol (THC), cannabidiol (CBD), cannabigerol (CBG), cannabichromene (CBC), cannabinol (CBN), tetrahydrocannabivarin (THCV). The common terpenes were bisabolol, camphene, β-Caryophyllene, α-humulene, limonene, linalool, β-Myrcene; cis- and trans-nerolidol; α-, β-, cis-, and trans-ocimene; α-pinene, β-pinene; α-terpinene, γ-terpinene, and terpinolene.

In the case of polar plots used to describe basic terpene profiles, α-pinene and β-pinene were summed together and shown as "pinene" (see Figs 7D–7F and 8D). For certain terpenes (ocimene and nerolidol), some labs measured individual isomers, and some reported a single total sum. In our main analyses using data aggregated across labs, we summed across cis- and trans-nerolidol, and across α-, β-, cis-, and trans-ocimene.

**Sample- vs. product-level analysis.** Most of the analysis was conducted on the sample-level, meaning the data analyzed were the individual *Cannabis* flower samples labs received and measured. We conducted some analyses at the product-level. A product represents the average cannabinoid and terpene measurements for all strain name-anonymized producer combinations. For example, Producer 101 might have 15 separate samples attached to the name "blue-dream" that were submitted over some period of time. For product-level analyses (Figs 5E, 5F, 7D-7F, 8A–8E, and 9A-9D), we averaged across such samples for each unique combination of Producer IDs and strain names. THC:CBD chemotype was assigned to products based on the average total THC and CBD values.

**Statistics.** When performing statistical tests, we opted for statistical tests that do not depend on assumptions about the distribution of the underlying data. For comparing groups, we used the Welch's t-test, which does not assume equal population variances. For correlations, we computed Spearman's rank correlation coefficient by default, as it provides a non-parametric measure of correlation. Any samples with null values among the variables being analyzed were excluded in the calculation. Significance levels were corrected using the most conservative Bonferroni correction to adjust for multiple comparisons, when applicable. All p-values reported in the figures and text as significant are significant at the corrected alpha level. Stars in figures (*, **, ***) correspond to the alpha levels 0.01, 0.001, and 0.0001 (with Bonferroni correction), respectively. Due to the large sample sizes in our dataset, we tended to obtain very small p-values that vary by many orders of magnitude. In these cases, p-values are reported as < 0.0001 (with Bonferroni correction).

With sufficiently large sample sizes, statistically significant p-values can be found even when differences are negligible. For this reason, we report effect sizes in addition to the p-values obtained from Welch's t-test. We used an adjusted version of Cohen's d ("d-prime") to estimate the effect size for independent samples without the assumption of equal variances [84]. This version averages the two population variances:

$$d\prime = \frac{X_1 - X_2}{\sqrt{\frac{\sigma_1^2 + \sigma_2^2}{2}}}$$

**Fig 1.** The total levels for the six common cannabinoids were visualized as combination violin and box plots. A scatter plot and a histogram of the relationship between total THC and total CBD were visualized with the THC:CBD chemotypes color-coded. Principal component analysis (PCA) was run on the normalized values of the six common cannabinoids (i.e., the % of measured common cannabinoids). Null values were filled with zeros. A PCA biplot was created to visualize the PCA scores of the samples and the weight of each cannabinoid on the first two principal components.

**Fig 2.** The data was filtered by each of the three chemotype classes identified in Fig 1 (THC-dominant, CBD-dominant, and balanced THC:CBD). Pairwise scatterplots for each permutation of the three most abundant cannabinoids (THC, CBD, CBG) were made for the three THC:CBD chemotype classes. Outliers were removed using the median absolute deviation method with a conservative threshold of 3.5 standard deviations. The resulting nine plots are visualized in Fig 2. The Spearman rank correlation for each cannabinoid relationship in each class was computed to measure the strength of the relationship. Statistical significance was evaluated after using the Bonferroni correction for 9 multiple comparisons. All observed relationships were significant at the (corrected) $P < 0.0001$ level.

**Fig 3.** The fourteen common terpenes were visualized for samples with terpene data in a combination violin/box plot, ordered by median value, descending. The linear relationships between two pairs of terpenes (α- and β-pinene, and β-caryophyllene and humulene) were quantified with a linear regression and Spearman rank correlation. Statistical significance was evaluated after using the Bonferroni correction for two multiple comparisons.

**Fig 4.** The fourteen terpene levels were correlated with each other using a Spearman rank correlation. A cluster map visualization in Fig 4 combining a heatmap and hierarchical clustering visualizations was made. Because of the multiple pairwise comparisons (14 x 13 / 2 = 91), statistical significance was evaluated after using the Bonferroni correction for 91 multiple comparisons. Cells were colored by the strength of the relationship (bluer are stronger negative correlations, redder are stronger positive correlations) and annotated with the correlation value only if the relationship was significant at the (corrected) $P < 0.05$ level. Only four compound combinations had non-significant corrected relationships: (1) terpinolene-nerolidol, (2) terpinolene-humulene, (3) myrcene-bisabolol, and (4) ocimene-camphene. The distances between clusters were evaluated using the "average" method in the "hierarchy.linkage" function and the "euclidean" function was used as a distance metric.

The clusters recovered by the clustermap visualization can also be represented as a network where the nodes are the terpenes and the (weighted) edges are the correlations. Because nearly all compound combinations have statistically significant correlations (even after Bonferroni correction), the resulting network would be (nearly) completely connected. To sparsify the network for visualization purposes, the correlation values were thresholded to greater than or equal to 0.10 to show the strongest relationships. There were 38 remaining edges after this thresholding procedure. This threshold value was chosen through qualitative iteration to generate a network that preserves all 14 compounds but is sufficiently sparse to visually recover the clusters identified in Fig 4A. The network was visualized using a spring-embedding layout algorithm and visualized using the "networkx" library in Python.

**Fig 5.** Principal component analysis (PCA) was run on the normalized values of the fourteen common terpenes (i.e., the % of measured common terpenes) on all samples with terpene data. Null values were filled with zeros (nearly identical results are seen by replacing these with the LLOQ for each lab). A bar plot was created to visualize how much variation each principal component captured in the data. PCA biplots were created to visualize the PCA scores of the samples and the weight of each terpene on the first three principal components (Fig 5B–5D).

Sample level data was averaged across strain name/producer ID pairs to create a product level dataset. Pairwise cosine distances of terpene profiles were calculated for products in each chemotype. We then averaged the cosine distances across each product, so each product had an associated average cosine distance. These values were plotted in a violin/box plot (Fig 5E). Welch's t-tests and effect sizes were calculated between each chemotype. Statistical significance was evaluated after using the Bonferroni correction for three multiple comparisons. The top terpene among the 14 common terpenes was found for each product. If the most abundant

terpene was not either myrcene, caryophyllene, limonene, terpinolene, alpha pinene, or oci-mene, the top terpene was listed as "other" (Fig 5F).

**Fig 6.** For Fig 6A–6F, the sample level data was filtered to include only THC-dominant samples with terpene data. Terpene data were normalized to be % of measured common ter-penes. Null values were filled with zeros. PCA was run on these normalized values and then plotted.

Silhouette coefficients for each sample were calculated using the mean nearest-cluster Euclidean distance (b) minus the mean intra-cluster Euclidean distance (a), divided by max (a, b). This value measures how similar a sample is to its labeled cluster compared to other clus-ters. The individual silhouette sample scores plotted were obtained from a random subsample of the data ($n = 10,000$) due to graphic memory limitations, however the average silhouette score displayed on the figure was obtained using the full filtered dataset.

We used the k-means clustering algorithm to segment THC-dominant samples based on terpene profiles. To determine the optimal number of clusters we created an 'elbow plot', which plots a range of number of clusters versus within-cluster sum of squared errors (S5A Fig). This revealed that the optimal number of clusters to use was k = 3. K-means clustering was applied to the normalized dataset. A color palette was created using the color of the most abundant terpene for each cluster's average terpene profile. The correct choice of k can be ambiguous, so we also explored our cluster analysis for k = 2 and k = 4 clusters (S5B and S5C Fig).

**Fig 7.** To evaluate the difference between the labeling methods described above, silhouette scores (described above) were calculated on the full dataset for the three different methods. Welch's t-tests and effect sizes were calculated between these methods. Statistical significance was evaluated after using the Bonferroni correction for three multiple comparisons.

A UMAP embedding [85] was run on the terpene data of THC-dominant samples and color coded by k-means cluster label. The parameters for number of components and number of neighbors were specified as 2 and 15, respectively. An interactive 3-D version of a similar product-level UMAP can be found here: https://plotly.com/~cj.smith015/5/. Each data point can be hovered over to reveal the following information: strain name, Indica/Hybrid/Sativa label, THC and CBD concentration, dominant terpene, and k-means cluster label information.

To illustrate a simple terpene profile, we ran k-means clustering (k = 3) on the product-level dataset. α- and β-pinene were summed together. The normalized terpene values and total THC, CBD, and CBG values from the THC-dominant product dataset were grouped by k-means cluster label and averaged. Polar plots were constructed based on the average terpene profiles and limited to eight terpenes to help with visual legibility. The terpene profiles of the top 25 products in each cluster with the most samples were drawn in grey behind the cluster-level average.

**Fig 8.** To quantify consistency between products attached with the same name we needed to ensure that the underlying data contained multiple samples per producer ID and several unique producer IDs each. We used the following thresholds: to be included, a strain name must be linked to at least five producers with at least five samples from each producer. If the strain met this threshold, we included all samples of that strain in our examination, averaging all samples linked to each unique producer ID to create product averages. 41 strain names met this threshold. Due to the predominance of THC-dominant samples in the dataset, all strain names in the list happened to be THC-dominant. Measures of "strain name" popularity were supplied in the form of normalized values for the number of unique views each page of this popular, public cannabis database.

In Fig 8B, a correlation matrix was constructed on the terpene values of THC-dominant samples for the ten strain names attached to the most samples. The samples were put in

descending order based on the number of samples, and within each strain name, ordered by producer ID. Pairwise cosine similarity scores were calculated on the samples and plotted as a heat map with a Gaussian filter for visualization purposes.

Cosine similarities were calculated for the terpene profiles of products for each strain name, then averaged to assign a mean similarity score to each product (identity values of 1 were replaced with nulls so as to not artificially increase the average). A violin/box plot was created with these similarity scores, ordered by median value. The dashed line in Fig 8C represents the average similarity score one would expect if strain names were randomly assigned, obtained by running a bootstrap simulation where strain names were shuffled across the product IDs. Average similarity scores for products were calculated based on these randomized strain names. Those scores were then averaged to give each (randomized) strain name a similarity score. A weighted average was created by taking the randomized strain-level similarity scores and weighing them by the number of products associated with each randomized strain name. This process was repeated 200 times and the mean of this distribution was calculated and displayed as the dashed line. Welch's t-tests and effect sizes were calculated comparing the similarity scores for each strain to the bootstrapped distribution of average randomized strain-level similarity scores. Statistical significance was evaluated after using the Bonferroni correction for 41 multiple comparisons.

A UMAP embedding was run on the normalized terpene data of the entire THC-dominant product dataset and color coded by k-means cluster label, k = 3. The parameters for number of components and number of neighbors were specified as 2 and 15, respectively.

**Fig 9**. Using the THC-dominant product dataset with k-means clustering (k = 3), a UMAP embedding was run on the normalized terpene data and color coded by Indica/Sativa/Hybrid labels.

Excluding products without an associated Indica/Sativa/Hybrid label, the percentage of Indica/Sativa/Hybrid labels for products was found for each k-means cluster label. Chi-squared tests were calculated comparing these percentages with the overall percentages. Statistical significance was evaluated after using the Bonferroni correction for three multiple comparisons.

Using the list of 41 strains obtained by the thresholds described for Fig 8, the most frequent k-means cluster label was identified for each strain name. The number of products with that cluster label divided by the total number of products for that strain multiplied by 100 gave the percentage of products in the top cluster. Up to seven strains in each cluster were displayed in the bar chart in Fig 9D, ordered by k-means cluster label and then by the percentage of products in the top cluster. The dashed line in Fig 9D represents the average percentage of products one would expect if strain names were randomly assigned, obtained by running a bootstrap simulation where strain names were shuffled across the product dataset, as described above for Fig 8. Welch's t-tests and effect sizes were calculated by comparing the distribution of products in the top cluster for each strain to the bootstrapped distribution of average percentage of randomized products in the top cluster. Statistical significance was evaluated after using the Bonferroni correction for 41 multiple comparisons.

## Supporting information

**S1 Fig. Violin plot of distribution of all cannabinoids measured, by region.**
(TIF)

**S2 Fig. Total THC vs. Total CBD levels, by region.**
(TIF)

**S3 Fig. Scatterplots showing the correlation between α- and β-pinene, by region.**
***P < 0.0001.
(TIF)

**S4 Fig. Scatterplots showing the correlation between β-caryophyllene and humulene, by region.** ***P < 0.0001.
(TIF)

**S5 Fig. (A)** Line plot showing the relationship between number of clusters in k-means clustering and within-cluster sum of squared errors, using THC-dominant sample terpene data. "Elbow point" was determined to be at k = 3. **(B)** PCA scores for all THC-dominant samples plotted along PC1 and PC2, color-coded by k-means cluster labels, k = 2. **(C)** PCA scores for all THC-dominant samples plotted along PC1 and PC2, color-coded by k-means cluster labels, k = 4.
(TIF)

**S6 Fig. PCA scores for THC-dominant samples plotted along PC1 and PC2, color-coded by k-means cluster labels attached to each sample, by region.**
(TIF)

## Acknowledgments

We thank Dr. Alex Wiltschko and Dr. Michael Tagen for helpful comments on the manuscript. We also thank representatives from each testing lab who agreed to allow us to conduct our study: CannTest (Alaska), ChemHistory (Oregon), Confidence Analytics (Washington), Modern Canna Labs (Florida), PSI Labs (Michigan), SC Labs (California).

## Author Contributions

**Conceptualization:** Nick Jikomes.

**Data curation:** Christiana J. Smith, Nick Jikomes.

**Formal analysis:** Christiana J. Smith, Brian Keegan.

**Investigation:** Nick Jikomes.

**Methodology:** Christiana J. Smith, Nick Jikomes.

**Project administration:** Christiana J. Smith, Nick Jikomes.

**Software:** Christiana J. Smith, Brian Keegan.

**Supervision:** Nick Jikomes.

**Validation:** Nick Jikomes.

**Visualization:** Christiana J. Smith, Daniela Vergara, Brian Keegan.

**Writing – original draft:** Christiana J. Smith, Daniela Vergara, Brian Keegan, Nick Jikomes.

**Writing – review & editing:** Christiana J. Smith, Daniela Vergara, Brian Keegan, Nick Jikomes.

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
