## [Decision Letter · Decision Letter 0]

7 Jan 2022

PONE-D-21-34155The phytochemical diversity of commercial *Cannabis* in the United StatesPLOS ONE

Dear Dr. Jikomes,

Thank you for submitting your manuscript to PLOS ONE. After careful consideration, we feel that it has merit but does not fully meet PLOS ONE’s publication criteria as it currently stands. Therefore, we invite you to submit a revised version of the manuscript that addresses the points raised during the review process.

We look forward to receiving your revised manuscript.

Kind regards,

MQ Shahid

Academic Editor

PLOS ONE

Journal Requirements:

Additional Editor Comments (if provided):

The results of the manuscript are interesting, however, reviewers have raised several concerns about the manuscript that must be clarified, especially about terminology used in the manuscript and lack of discussion. So, this manuscript required a major revision before considering for publication. Please see attached review reports for detail.

Reviewers' comments:

Reviewer's Responses to Questions

**Comments to the Author**

1. Is the manuscript technically sound, and do the data support the conclusions?

Reviewer #1: Partly

Reviewer #2: Partly

Reviewer #3: Yes

2. Has the statistical analysis been performed appropriately and rigorously? 

Reviewer #1: I Don't Know

Reviewer #2: No

Reviewer #3: Yes

3. Have the authors made all data underlying the findings in their manuscript fully available?

Reviewer #1: Yes

Reviewer #2: Yes

Reviewer #3: Yes

4. Is the manuscript presented in an intelligible fashion and written in standard English?

Reviewer #1: Yes

Reviewer #2: Yes

Reviewer #3: Yes

5. Review Comments to the Author

Reviewer #1: 1. Several issues concerning the material and method section are of concern:

A. ANALYTICAL PROCEDURES: The material and methods section should be revised to include a section about analytical procedures. Detailed analytical procedures of the chemical analyses procedures must be included as is acceptable for scientific papers. The very general description given so far mainly for instrumentations is insufficient.

B. Results from several laboratories using unknown analytical procedures of varied instrumentations were pulled for the study. This is a very large disadvantage, since large variations in chemical analysis results for cannabinoids and terpenes are known to results from analysis at various laboratories; and no standardization was conducted in the present study. It can therefore not be excluded that variabilities reported in the study, for example regional variability, do not source from variations in accuracy of the analytical techniques. For this affect, the detailed analytical procedures used in each laboratory should be detailed in the manuscript (as is mentioned in point A above). Furthermore, the authors should consider to use reference material to be analyzed by all laboratories, and to normalize the data used for the study based on identified inconsistencies between laboratories, prior to the data evaluation conducted in the study.

C. THE SOURCE OF THE SAMPLED MATERIAL: A proper description of the source of the material used for the study is required. Are these samples from dispensaries of marketed products? From growers? From consumers? Etc. I suppose that these are results of the commercial labs datasets. More detailed information about the source of the samples should be supplied.

2. Another problematic issue that should be addressed is that some text parts reads like a commercial/advertising in a form that is not suitable for scientific publications. Specifically: A. The first paragraph of the results section: You can mention the source of the data ones in the manuscript, and all other text about Leafly (this also includes line 660-662 ) and all other mentions of leafy in the manuscript such as "leafy supplied.., leafy included… " should be removed.

B. Line 704-723: This section and the details of the lab are of commercial nature, and have no place in a M&M section of a scientific report. It should be removed from the manuscript. Relevant information pertaining to the M&M can be added to the text. Some credit, albeit not in the form it is now written can be added to the acknowledgment section.

3. Scientific background: The introduction lacks a discussion about potential reasons other then genetics diversity between strains for the variability in the secondary metabolite profile between samples. There is a growing body of scientific information about effects of cultivation conditions such as mineral nutrition (N, P, K, ), light, and plant architecture on secondary metabolite production in cannabis.

4. Terminology: The use of the word cannabis in the manuscript to describe cannabis derived products as well as cannabis plant material products and the cannabis plant is misleading and confusing. It is confused with concentration in the cannabis plant. Therefore them manuscript should be revised throughout for terminology, to distinct between cannabis plant material, and cannabis derived products. One example out of many for this issue is line 101 – it is unclear whether this sentence refers to differences in cultivation in one area, or products distributed in one area.

5. Abstract: remove the words "After careful descriptive" from line 32. Just describe your results and conclusions. All scientific work should be carefully conducted.

6. Line 35-36. "However, certain labels are statistically overrepresented for specific chemotypes" - This is not clear.

7. The last sentence of the abstract: it is not clear from the abstract why this is the conclusion of the study.

8. Line 52-53: remove the uncommon names, and stay focused in the introduction on the topic of study.

9. The sentence in lines 54-56 is not supported by references.

10. Throughout the manuscript, Δ-9-tetrahydrocannabinolic should be corrected for the proper writing.

11. Paragraph starting in line 70: this paragraph is written in a popular style of writing. It needs to be revised to reflect scientific writing style for chemical pathways and processes.

12. The paragraph in lines 119-134 is to long. The discussion concerning medical effect of the metabolites interactions should be reduced considerably.

13. The lists in pages 32-33: should be converted to regular text in sentences (not lists).

14. I am not an expert of the statistical methods used and the curve fitting techniques used. But judging the appearance of the regression lines in figure 2, it seems that for many of the sub-figures that regression lines are extrapolations and does not visually fit the trend for the bulk majority of the data. These curve fits should be reevaluated.

15. Figure 3: because a uniform scale was selected for the Y axis for all terpenes, it is impossible to see the trends for most (minor) terpenes.

Reviewer #2: PONE-D-21-34155

The phytochemical diversity of commercial *Cannabis* in the United States

Smith et al.

The manuscript The phytochemical diversity of commercial Cannabis in the United States reports the results of an analysis of aggregated cannabinoid and terpene plant secondary metabolite data in the species Cannabis sativa. The authors interpret this data thoughtfully and provide new illumination on several standing topics. Additionally, the authors propose a helpful model to help explain the chemotaxonomic chemical relationships that help explain patterns seen within this dataset.

Overall, this work is compelling and adds tangible value to the larger discussion of several critical topics in Cannabis research. For the most part, the statistical approach is strong but requires additional scrutiny before publication. The authors should be praised for making both the underlying dataset and python code easily accessible online.

The introduction of seems to lack an understanding of the underlying genetic pathways that control biosynthesis of cannabinoids and terpenes in cannabis. This is important information to interpret their findings and is largely neglected. A more comprehensive literature review of cannabis genetics and biosynthetic pathways would greatly strengthen the value of this work.

I recommend that this paper undergo major revisions before resubmission.

There are several specific elements that if improved would make this work considerable stronger.

1. The terminology used throughout this paper often feels muddled or inconsistent. For example, the terms “sample”, “strain”, “variety” are not well defined. To unify cannabis research with the larger field of plant science, this paper reinforces the meaningless term "strain" as valid nomenclature for cannabis cultivars. This usage is unhelpful and should be deprecated. There are more precise and specific terms e.g., cultivar, population, selection, hybrid, landrace that would bring more clarity to this work and not muddle scientific research.

2. The lack of discussion and analysis of sample duplication or potential duplications is troubling. The manuscript claims that the study uses 90 thousand samples is misleading. In fact, the paper uses repeated measurements of only 3,439 unique named cultivars but does not state this. There should be much more attention paid to statements such as “The final number of samples was 89,923” which easily misleads the casual reader to think that this study is capturing 25 times more cultivars than it is. For example, there are over 1000 “samples” of ‘blue-dream’, ‘original-glue’, ‘dutch-treat’, and ‘gsc’ each. In the raw dataset, 481 cultivars have only one observation. This is inconsistency is not addressed and should be taken very seriously. Please report the number of cultivars and measures of dispersion in the abstract and introduction.

3. The authors of this manuscript do not seem to fully grasp that the mechanism of THC:CBD ratio is well-defined from a genetics perspective, e.g. the genotype at the B locus encodes THCA synthase or CBDA synthase. The term “Balanced THC:CBD” simply reflects a heterozygous individual at this locus. Please see the work of de Meijer and others who have clearly defined this locus (https://doi.org/10.1093/genetics/163.1.335 or https://genome.cshlp.org/content/29/1/146.short). Additionally, the authors should consider carefully how to use the term “chemotype”. In most cannabis research, the term “chemotype” is used in a discrete and specific way (e.g., “Chemotypes I through V”), although in this paper, the authors tend to switch back and forth between using this term in an informal way to discuss any secondary metabolites. This will be confusing to many readers.

4. The authors often replace missing data with zeros (e.g., line 854). I am not at all confident that this is the appropriate statistical treatment.

5. The authors do not correctly state the legal definition of hemp as total THC in line 83. See 2021 Final Rule https://www.federalregister.gov/documents/2021/01/19/2021-00967/establishment-of-a-domestic-hemp-production-program.

6. In line 83, the authors mention that the 0.3% threshold was “arbitrarily defined”. This is not precise; the threshold was largely based on previous work by Ernest Small (see https://www.jstor.org/stable/4253607) who thought at the time that the upper limit of THC in hemp would always be less that 0.3%.

7. The authors should consider including the biochemical pathways of cannabinoid and terpene biosynthesis and their molecular structures. Proving a prior on how these metabolites are synthesized would be an illuminating compliment. For example, a clearer model of cannabinoid biosynthesis would help clarify that the variation observed in this data is likely a result of several severe genetic bottlenecks / founder effects / selective sweeps within the larger pool of cannabis germplasm, rather than random variation that might be observed within natural populations.

8. Figure 10 is very useful and presents a novel model to help explain metabolic variation in cannabis germplasm. The authors might consider using a more quantitative visualization of this data, rather than representing relationships with line width. For example, a Sankey diagram might be more appropriate to show this relationship more quantitatively (see https://en.wikipedia.org/wiki/Sankey_diagram).

9. In line 577, the authors state that “Marketing emphasizing Indica-labelled products as sedating and Sativa-labelled products as energizing are not borne out by our analysis of the underlying chemistry”. I personally believe that the authors assumptions are correct, but I am not confident that they can make this claim given the evidence that they present.

10. The authors gloss over that there appears to be no representation of hemp within this study. Some clarification or connection to work that has evaluated hemp secondary metabolites would be helpful.

11. The authors exclude 0% cannabinoids from their analysis. This is probably correct, although cannabinoid knockout material does exist.

12. The correlations and trendlines in Figure 2 look very strange, especially the “Balanced” comparison between total THC and total CBD. Please verify that this analysis is correct and appropriate.

Reviewer #3: I found the paper to be well written and constructed. The intro could perhaps benefit from a reduction in length. The results section also at times blends commentary that could be moved to the discussion or methods sections. The discussion section really ties together the major findings of the analyses and highlights the importance of the work.

I think an important analytical consideration to discuss is the magnitude of the significant differences for the groups, especially regarding the terpenes. While the clustering analyses are compelling, what are these significant differences really telling us about the samples? The terpenes are measured at a few percent at most, so how big are the significant differences between the groups in terms of terpene levels? I would also like to read some additional discussion on any outliers that stand out.

I recommend not using the term legal cannabis industry. Most state laws use the term marijuana, despite the historical baggage of that term. Either way, I think it should be clarified what is known about these the origins of these samples and how sample selections may have affected the results. While the methods describe the relationship between the testing labs the Leafly to some extent, I don't see anything about the lab / cultivator relationship--not everyone knows about the state testing requirements etc. I assume these are state licensed marijuana cultivators' samples. This would exclude hemp industry samples, many of which are genetically very similar to MJ (Grassa et al. 2021).

Line number comments:

71: Varieties with > 15% CBGA are common in the US now. Same for THCV in some areas.

89: Include de Miejer breeding papers and terms?

252 – 254: I would say more recent, rather than historic. See for example how older and non US samples have significant CBD content:

Mikuriya, T. H., and Aldrich, M. R. (1988). Cannabis 1988 old drug, new dangers the potency question. Journal of Psychoactive Drugs 20, 47–56. doi:10.1080/02791072.1988.10524371.

Hanuš, L. O., Levy, R., De La Vega, D., Katz, L., Roman, M., and Tomíček, P. (2016). The main cannabinoids content in hashish samples seized in Israel and Czech Republic. Israel Journal of Plant Sciences 63, 182–190. doi:10.1080/07929978.2016.1177983.

275 – 286: These analyses reflect highly unbalanced sample numbers in the dataset for type I, II, and III plants.

6. PLOS authors have the option to publish the peer review history of their article (what does this mean?). If published, this will include your full peer review and any attached files.

Reviewer #1: No

Reviewer #2: **Yes: **Zachary James Stansell

Reviewer #3: No

---

## [Author Response · Author response to Decision Letter 0]

11 Mar 2022

Thank you very much for the kind words and for taking the time to review our manuscript. We address all comments raised by reviewers in our file 'Answer for Reviewers.docx' (labeled as Response to Reviewers Feb 2022).

We have also updated our cover letter to be clearer that the authors received no specific funding for this work. Leafly did not provide specific funding for this work beyond allowing Nick Jikomes to spend some of his professional time overseeing the project.

---

## [Decision Letter · Decision Letter 1]

11 Apr 2022

The phytochemical diversity of commercial *Cannabis* in the United States

PONE-D-21-34155R1

Dear Dr. Jikomes,

We’re pleased to inform you that your manuscript has been judged scientifically suitable for publication and will be formally accepted for publication once it meets all outstanding technical requirements.

Kind regards,

Muhammad Qasim Shahid

Academic Editor

PLOS ONE

Additional Editor Comments (optional):

Authors have done the suggested changes, so manuscript could be accepted for publication.

Reviewers' comments:

Reviewer's Responses to Questions

**Comments to the Author**

1. If the authors have adequately addressed your comments raised in a previous round of review and you feel that this manuscript is now acceptable for publication, you may indicate that here to bypass the “Comments to the Author” section, enter your conflict of interest statement in the “Confidential to Editor” section, and submit your "Accept" recommendation.

Reviewer #2: All comments have been addressed

Reviewer #3: All comments have been addressed

2. Is the manuscript technically sound, and do the data support the conclusions?

Reviewer #2: Yes

Reviewer #3: Yes

3. Has the statistical analysis been performed appropriately and rigorously? 

Reviewer #2: Yes

Reviewer #3: Yes

4. Have the authors made all data underlying the findings in their manuscript fully available?

Reviewer #2: Yes

Reviewer #3: Yes

5. Is the manuscript presented in an intelligible fashion and written in standard English?

Reviewer #2: Yes

Reviewer #3: Yes

6. Review Comments to the Author

Reviewer #2: Thank you for thoroughly addressing the reviewers comments-- I do believe it's considerably stronger given your revisions.

I can tell that this was a challenging paper to write but it does provide unique and compelling information-- thank you for your efforts.

Reviewer #3: I think this should be published as is. The manuscript and data clearly meet the PLOS ONE Criteria for Publication. Overall, the data and analyses generally recapitulate what has been published about strain names and the indica/sativa system, but this project uses many fold more samples than have been used to date. I think the specific clusters and groups that emerged from the analyses by Fig 6 could be useful for future work. Personally, I don't see a clear road forward for studying the endless chemical combinatorial options for cannabis in a clinical setting, but perhaps studies like this can at least simplify the landscape to some extent. Maybe that's not a realistic goal though and we should accept cannabis as a complex natural product. Whether or not cannabis nomenclature can or ever will be cleaned up and used in a consistent way that reflects chemical composition remains to be seen. In other plants, taste and aroma traits are also poorly understood (https://www.pnas.org/doi/abs/10.1073/pnas.2118676119), but notably are not solely driven by terpenoids, as is often claimed in cannabis.

7. PLOS authors have the option to publish the peer review history of their article (what does this mean?). If published, this will include your full peer review and any attached files.

Reviewer #2: No

Reviewer #3: No

---

## [Editor Report · Acceptance letter]

13 Apr 2022

PONE-D-21-34155R1 

The phytochemical diversity of commercial *Cannabis* in the United States 

Dear Dr. Jikomes:

I'm pleased to inform you that your manuscript has been deemed suitable for publication in PLOS ONE. Congratulations! Your manuscript is now with our production department. 

Kind regards, 

on behalf of

Dr. Muhammad Qasim Shahid 

Academic Editor

PLOS ONE